



# Objectively identified mesoscale surface air pressure waves in the context of winter storm environments and radar reflectivity features: a 3+ year analysis

Luke R. Allen[1], Sandra E. Yuter[1,2], Matthew A. Miller[2], and Laura M. Tomkins[1,3]

[1]Center for Geospatial Analytics, North Carolina State University, Raleigh, NC, 27695, USA
[2]Department of Marine, Earth, and Atmospheric Sciences, North Carolina State University, Raleigh, NC, 27695, USA
[3]Current affiliation: Karen Clark and Company, Boston, MA, 02116, USA

**Correspondence:** Luke R. Allen (lrallen3@ncsu.edu) and Sandra E. Yuter (seyuter@ncsu.edu)

**Abstract.** Atmospheric gravity waves (i.e., buoyancy waves) can occur within stable layers when vertical oscillations are triggered by localized heating, flow over terrain, or imbalances in upper level flow. Case studies of winter storms have associated gravity waves with heavier surface snowfall, but the representativeness of those findings for settings without orographic precipitation has not been previously addressed.

5     To detect gravity waves, we deployed networks of high precision pressure sensors from January 2020 to April 2023 in and around Toronto, ON, Canada, and New York, NY, USA, two regions without strong topographic forcing. Pressure wave events were identified when at least 4 sensors in a network detected propagating pressure waves with wave periods $\leq 67$ min, wavelengths $\leq 170$ km, and amplitudes $\geq 0.45$ hPa. We detected 33 pressure wave events across 40 months of data, of which 23 were gravity waves and the rest were frontal passages, outflow boundary passages, or a wake low.

10     Reanalysis model output and operational weather observations provided environmental context for each gravity wave event. Consistent with previous work, most gravity wave events occurred with strong upper-level flow imbalance to the south or west of their location. Of the 79 winter storms with snow that occurred over our 40 months of observations, only 6 had detectable gravity waves. For New York City, the typical offshore cyclone low center track means the metro area is usually in a location where gravity waves are not expected to occur.

## 1 Introduction

Atmospheric gravity waves (i.e., buoyancy waves) result from the forced perturbation of air parcels in a statically stable environment such that parcels oscillate about their original height with parcel buoyancy acting as the restoring force (Nappo, 2002). Possible triggering mechanisms for gravity waves can include, but are not limited to, forced flow over topography, adjustment to imbalanced flow, and localized latent heating (Fritts and Alexander, 2003). The resulting up and down motions 20 can propagate outwards from the location of their originating triggering mechanism through stable layers, but do not yield net movement of fluid, nor do they necessarily follow the mean flow of the air through which they propagate.





Case studies of winter storms have associated gravity waves with heavier surface snowfall but the representativeness of these findings for settings without strong topographic forcing has not been systematically examined. It is the goal of this paper to remedy that gap with a comprehensive analysis of observed gravity waves based on 40 months of pressure sensor network data. The waves we present in this paper are propagating (i.e., not terrain-locked), moderate to high-amplitude (up to 5.5 hPa, in a similar range to Uccellini and Koch, 1987), and have short to moderate wavelengths (up to 170 km) and wave periods (up to 67 min). The types of disturbances which were detected by our networks of pressure sensors included not only gravity waves, but also synoptic fronts, convective outflow boundaries, and a convective wake low. We will describe how we discerned those four types of disturbances from one another in Sects. 2.2-2.3. In order to keep terminology clear and consistent, and because different studies may use different conventions, Table 1 defines terms used in this paper related both to wave properties and the different types of waves.

Many observational studies have used pressure sensors to detect gravity wave signals (e.g., Christie et al., 1978; Kjelaas et al., 1974; Einaudi et al., 1989; Uccellini and Koch, 1987; Bosart et al., 1998; Grivet-Talocia et al., 1999; Koch and Siedlarz, 1999). In this study we use networks of high precision pressure sensors to detect and track gravity waves and analyze them in the context of radar-detected features in winter storms. Table 2 compares the properties of the waves presented in this paper to those in a selection of other papers in the literature. The spatial and time scales of pressure waves that we focus on, 3.5 to 170 km and 2 to 67 min overlap the larger end of the scales examined in previous work by Christie et al. (1978) and overlap the smaller end of the scales examined by Grivet-Talocia et al. (1999) and Uccellini and Koch (1987).

## 1.1 Possible effects of gravity waves on cloud and precipitation processes

Gravity waves can have noticeable effects on cloud and precipitation processes when they occur within the troposphere and under the right conditions. In marine stratocumulus, upward motions associated with gravity waves can yield enhancements in drizzle (Allen et al., 2013; Connolly et al., 2013). In the southeast Atlantic, satellite observations have revealed cases of marine stratocumulus cloud decks rapidly eroding, and this abrupt cloud-clearing appears to be related at least in part to gravity waves (Yuter et al., 2018; Tomkins et al., 2021). In deep convection, latent heating can trigger gravity waves of varying frequency which alter the pre-storm environment and lead to initiation of convection ahead of the existing line of storms, often referred to as "action at a distance" (Fovell et al., 2006; Adams-Selin, 2020) and "gregarious convection" (Nicholls et al., 1991; Mapes, 1993; McAnelly et al., 1997). There has been extensive work describing mechanisms whereby terrain-locked gravity waves can enhance clouds and precipitation (e.g., Gaffin et al., 2003; Colle, 2004; Doyle and Durran, 2007; Houze, 2012, 2014; Kingsmill et al., 2016; Ma et al., 2023).

In order for gravity waves to modify clouds and precipitation, several processes have to occur within suitable conditions in sequence (Fig. 1). Gravity waves are first triggered, then the waves propagate away from their source location and may be ducted. If cloud is not already present in the wave duct, then the upward branches of the gravity waves must lift parcels to saturation, either with respect to ice (for air temperature $< 0°C$) or with respect to liquid water (for air temperature $\geq 0°C$), in order for cloud to form. If conditions are saturated or supersaturated in the wave duct (either $RH_{ice} \geq 100\%$ and/or $RH_{water} \geq 100\%$), then enhanced vapor deposition and/or condensation can occur in the upward branches of gravity waves. If lifting associated





with a gravity wave brings an ice or mixed phase cloud parcel to liquid water saturation, and riming then occurs, that rimed ice mass will not be removed unless $RH_{ice}$ falls below 100% in the downward branch of the gravity wave and sublimation occurs. When net increases in particle mass due to gravity waves are sufficient to enlarge cloud particles to precipitation-size, the precipitation that falls out of the parcel results in a net loss of total water from the parcel which is not reversible (e.g., Allen
et al., 2013).

## 1.2  Gravity waves in winter storm case studies

Previous case studies of gravity waves in winter storms include those that are close to the scale range targeted by this study (wavelengths $\leq$ 170 km and wave periods $\leq$ 67 min) and those that examined larger scale phenomena. Gaffin et al. (2003) described a heavy snowfall event where gravity waves generated by flow over terrain in the lee of the Smoky Mountains
contributed to localized lifting. Bosart et al. (1998) presented a case of a very large-amplitude gravity wave (with a peak-to-trough pressure difference on the order of 10 hPa) associated with observed snowfall rates up to 15 $\mathrm{cm\,hr^{-1}}$ (this refers to snow depth, **not** liquid equivalent) in the northeastern United States on 4 January 1994. The gravity wave in this case propagated at roughly 30-40 $\mathrm{m\,s^{-1}}$ toward the northeast with a wavelength of 200-300 km (implying a wave period of roughly 1.4-2.8 h). Zhang et al. (2001) simulated the 4 January 1994 case presented by Bosart et al. (1998) using the National Center for
Atmospheric Research/Pennsylvania State University Mesoscale Model 5. Their analysis of the simulated case indicated that geostrophic adjustment in the exit region of an upper-level jet streak initially triggered lower-amplitude gravity waves (roughly 1 hPa peak-to-trough), which merged and had a resonant interaction with an upper-level front leading to their nonlinear amplification. Zhang (2004) later generalized the term "geostrophic adjustment" as "balance adjustment" for curved flows as the trigger for gravity wave genesis. In the model results shown by Zhang et al. (2001), the upward motion associated with
the interaction of the gravity wave and the upper-level front led to release of potential instability and a region of elevated convection, where heavy precipitation was produced in the simulation.

The conditions necessary for gravity wave generation by balance adjustment often exist in the strong baroclinic trough-ridge systems which produce winter storms. After gravity waves are generated aloft, their energy propagates upward and downward. If appropriate conditions exist, the waves can then be *ducted* or trapped within a cloud layer, allowing the waves to influence
cloud processes (Ruppert et al., 2022). Lindzen and Tung (1976) described the theoretical conditions for an ideal wave duct: an absolutely stable ducting layer (where the environmental lapse rate < the moist adiabatic lapse rate) beneath a statically neutral or conditionally unstable reflecting layer (where the environmental lapse rate $\geq$ the moist adiabatic lapse rate). Such lapse rate conditions are common ahead of a warm/stationary front or behind a cold front (e.g., Uccellini and Koch, 1987).

## 1.3  Reflectivity bands and velocity waves in winter storms

Gravity waves have been previously suggested as the key mechanism yielding locally enhanced bands of radar reflectivity in snow and Doppler velocity waves. Detection of wave signals using an array of pressure sensors can help distinguish gravity waves from the other candidate processes (Nappo, 2002, Sect. 8.2).



Linear regions of locally enhanced radar reflectivity (*bands*) are frequently observed in winter storms (e.g., Novak et al., 2004; Hoban, 2016; Ganetis et al., 2018). These bands are conventionally categorized into two types: *primary band* and sets
of *multibands*. Winter storms have been found to contain both a primary band and multibands, only a primary band, only multibands, or no bands at all (Hoban, 2016; Ganetis et al., 2018). Primary bands are $\geq 200\,\mathrm{km}$ long, usually 30-70 km wide, occur as a single feature in reflectivity within a given storm, and have been associated with regions of strong frontogenesis along an occlusion (Novak et al., 2004; Baxter and Schumacher, 2017; Ganetis et al., 2018). Multibands are $< 200\,\mathrm{km}$ long, usually 10-50 km wide, and occur in groups of two or more bands, which are often roughly evenly spaced (Hoban, 2016;
Ganetis et al., 2018). Given propagation speeds on the order of 10-30 $\mathrm{m\,s^{-1}}$, it can take enhanced reflectivity bands on the order of 5 min to 2 h to cross a given location.

Ganetis et al. (2018) found no robust correspondence between frontogenesis and the occurrence of multibands. Processes which may lead to multibands include gravity waves (Gaffin et al., 2003; Hoban, 2016), Kelvin-Helmholtz waves (Houser and Bluestein, 2011), shear-organized lines of cloud-top generating cells (Keeler et al., 2016, 2017), and convective cells elongated
by flow anomalies resulting from potential vorticity dipoles (Leonardo and Colle, 2023). A further complication in relating enhanced reflectivity in snow to heavy snowfall rates is that reflectivity can be increased by aggregation or partial melting of ice particles, which would not increase the associated snow mass. Additionally, localized reflectivity enhancements observed by radar a few km above the surface may not reach the surface (Tomkins, 2024).

Hoban (2016) and Miller et al. (2022) identified waves in the Doppler velocities ("Doppler velocity waves") measured
by National Weather Service NEXRAD radars (WSR-88Ds) in the Northeast United States during winter storms using the difference field between successive radar scans. Hoban (2016) analyzed 71 winter storms which contained multibands. Of those 71 storms with multibands, 50 also contained coherent sets of propagating Doppler velocity waves. If the sets of propagating parallel Doppler velocity bands have a surface pressure signal, they could be gravity waves. If not, other mechanisms such as Kelvin-Helmhotz waves are a more likely.

## 110  1.4 Objectives of this study

In this study, we used high-precision surface pressure sensors to objectively identify pressure wave events over a 3+ year period, characterize the wave properties and their synoptic environments, and examine if the pressure waves are related to enhancements in radar reflectivity and coherent sets of Doppler velocity waves. Section 2 describes the pressure data and processing techniques to extract wave events, the reanalysis model output used to characterize the large scale environment, and
the several types of observations used to identify enhanced reflectivity bands, Doppler velocity waves, temperature inversions, and surface snow rates. Section 3 describes the characteristics of pressure wave events and their environmental context. Section 3.1.3 puts the pressure waves into context of radar-detected features, including how often they were collocated and moving with enhanced reflectivity bands and/or Doppler velocity waves. Finally, Sect. 4 includes conclusions and discussion of the results with potential avenues for future work.



## 2 Data and Methods

We used networks of pressure sensors in the Toronto, ON, Canada, and New York, NY, USA, metropolitan areas to detect wave events (Allen et al., 2024d, and Fig. 2), and we analyzed the context of those wave events using ERA5 reanalysis data (Hersbach et al., 2020), radiosonde data from the Integrated Global Radiosonde Archive (IGRA; NOAA National Centers for Environmental Information, 2021b), surface weather data from Automated Surface Observing Systems (ASOS; NOAA National Centers for Environmental Information, 2021a), and operational S-band radar data from US National Weather Service (NWS) WSR-88D radars (NEXRAD; NOAA National Weather Service Radar Operations Center, 1991).

### 2.1 Pressure sensor data

In numerical model output, gravity waves can be identified by analyzing the 3D gridded fields of pressure, geopotential height, wind, and temperature perturbation values. In observations, pressure sensor data are needed to definitively confirm the presence of gravity waves. We deployed high precision pressure sensor networks in the Toronto, ON, Canada, and New York, NY, USA, metropolitan areas over a three year period (sensor locations shown in Fig. 2). To minimize costs and hassle, these sensors were located in the homes and offices of our collaborators and automatically reported back to a server at North Carolina State University where the data were archived.

Each instrument utilized either a Bosch BME280 (Bosch, 2022) or a Bosch BMP388 (Bosch, 2020) pressure sensor, and the timestamps, data logging, and communications were handled by Raspberry Pi Zero single board computers (Allen et al., 2024d). Each sensor was placed indoors to minimize wind contamination in the pressure measurements. The noise floor of the sensors is roughly 0.8 Pa, depending on ambient conditions. The sensors continuously recorded pressure at 1-second intervals when possible, but power or internet outages occasionally caused gaps in the data record (Fig. 3). We analyzed data between January 2020 and April 2023. Most of the sensors in New York and Long Island were deployed prior to January 2020 while the sensors in Toronto were deployed starting in October 2020 (Fig. 3). Analysis subsequent to the deployment of the sensors suggests that the smaller spatial scale and more circular pattern of the Toronto network as compared to the larger spatial scale and more linear west-east arrangement of the sensors in New York likely makes the Toronto network better at detecting smaller amplitude pressure waves.

#### 2.1.1 Detection of wave events

Allen et al. (2024d) described the methods for detecting waves in the pressure sensor data in detail, which are summarized here. To smooth out artifacts and high-frequency pressure variations, we use 10-s samples of pressure in hPa (i.e., we take the 10-s moving average then use every 10th point of the smoothed time series). The detection method relies on a wavelet transform, a technique for identifying wave signals in time-wave period (or time-frequency) space, which is preferable to Fourier transforms for finding transient (i.e., time-localized) waves. We used an analytic morse wavelet (Olhede and Walden, 2002; Lilly and Olhede, 2012) and analyzed wave periods between 1 and 120 minutes to detect waves on similar temporal





scales to enhanced reflectivity bands. The output of the wavelet transform is an array of complex values in time-scale space. The absolute value of those values is referred to as the wavelet *power* ($|W(b,a)|$, units $\text{hPa}^2\,\text{s}^{-1}$).

A scale-dependent threshold function $A(a)$ was defined using the mean wavelet power across the full data set by scale, multiplied by a constant $K$:

$$A(a) = K\,\langle|W(b,a)|\rangle_b \tag{1}$$

To identify only the strongest wave signals, we used $K = 10$. Mean wavelet power increases with wave period (Allen et al., 2024d, their Fig. 5). Wave events were then identified as peaks in the wavelet power which exceeded $A(a)$, along with their connected regions which exceeded $\frac{A(a)}{2}$. We refined these regions using the watershed transform to separate distinct signals at different wave periods, then took the bounding box to obtain the final event regions. The wavelet transform was then inverted over the final event region to extract the wave event trace (Allen et al., 2024d).

After wave event traces were extracted for each sensor individually, we identified coherent wave events across multiple sensors using the cross-correlation function $C_{ij}(\Delta t)$:

$$C_{ij}(\Delta t) = \frac{1}{||p_i||\,||p_j||}\int p_i(t)p_j(t + \Delta t)\,dt \tag{2}$$

where $p_i(t)$ and $p_j(t)$ are the extracted wave event traces for two sensors $i$ and $j$. Events in pairs of sensors were matched together if the maximized $C_{ij}(\Delta t)$ value exceeded 0.65, with the time between wave passages at the two sensors estimated by the corresponding time lag $\Delta t_{opt}$ (in s). The cross-correlation function and associated $\Delta t_{opt}$ values were calculated for each possible pair of sensors within a network, which produced a vector of time lags $\boldsymbol{t}$ representing the time between wave passages at each pair of sensors which captured the event.

We then calculated the *slowness vector* using the time lags $\boldsymbol{t}$ for each wave event. The slowness vector is a two element vector $\boldsymbol{s} = (s_x,\ s_y)$, where $s_x$ and $s_y$ (in $\text{s}\,\text{m}^{-1}$) are the inverses of the x- and y-components of the wave phase velocity, $\frac{1}{c_x}$ and $\frac{1}{c_y}$ (in $\text{m}\,\text{s}^{-1}$), respectively. We solve for $\boldsymbol{s}$ starting from the following equation (Del Pezzo and Giudicepietro, 2002):

$$\boldsymbol{t} = \boldsymbol{s} \cdot \boldsymbol{\Delta x} \tag{3}$$

where $\boldsymbol{\Delta x}$ is the two-column matrix of the x- and y-components of the distance vectors (in m) between each pair of sensors which captured the event. For events captured by at least 3 sensors, Eq. 3 represents an overdetermined system of linear equations, from which $\boldsymbol{s}$ is estimated using a least-squares approach:

$$\boldsymbol{s} = (\boldsymbol{\Delta x}^T\boldsymbol{\Delta x})^{-1}\boldsymbol{\Delta x}^T\boldsymbol{t} \tag{4}$$

where superscript T indicates the transpose of a matrix (Del Pezzo and Giudicepietro, 2002). The components of the slowness vector are then inverted to obtain the wave phase velocity vector $\boldsymbol{c} = (c_x,\ c_y)$. We assessed this phase velocity estimate by calculating the "modeled" delay times $\boldsymbol{t_m}$ using Eq. 3 with the estimated slowness vector. We calculated the root mean square error (RMSE, in s) and normalized root mean square error (NRMSE, unitless) of the modeled delay times as follows:

$$RMSE = \sqrt{\frac{\sum_{i=1}^{N_s(N_s-1)/2}(t_{m,i} - t_i)^2}{N_s(N_s-1)/2}} \tag{5}$$



$$NRMSE = \sqrt{\frac{\sum_{i=1}^{N_s(N_s-1)/2}(t_{m,i}-t_i)^2}{\sum_{i=1}^{N_s(N_s-1)/2}(t_i)^2}} \tag{6}$$

where $N_s$ is the number of sensors which captured a given event.

To have reasonable confidence in the wave phase velocity estimate for a given event, we require the event to be captured by at least 4 sensors with RMSE below 90 s and NRMSE below 0.1, as discussed by Allen et al. (2024d). Additionally, the following results exclude any wave events found between 15 January and 18 January 2022, following the Hunga-Tonga volcanic eruption on 15 January 2022, which produced a Lamb wave with measurable pressure signals globally (Adam, 2022; Burt, 2022; Allen et al., 2024d). After applying those criteria, 33 total trackable pressure wave events were detected by the Toronto and New York pressure sensor networks between January 2020 and April 2023. Nineteen pressure wave events were detected by the Toronto pressure sensor network, and 14 were detected by the New York pressure sensor network (Table 3).

In general, pressure waves of amplitude above roughly 0.1 hPa could be detected in a single sensor, depending on the wave period (shorter waves had a lower detection threshold). However, the waves with lower amplitudes (i.e., weaker signals) were more difficult to track across multiple sensors. As a result, each of the trackable pressure waves in this study had an amplitude of at least 0.45 hPa. Table 2 compares this amplitude to other studies in the literature, which found pressure waves between 0.05 and 10 hPa in amplitude.

## 2.2 ERA5 reanalysis data

We used hourly ERA5 reanalysis data (Hersbach et al., 2020) to characterize the large-scale environment near the surface and in the upper-troposphere during each wave event. ERA5 data are output to a global, 0.25 degree grid on constant pressure and height levels. From ERA5 data, we calculated equivalent potential temperature ($\theta_e$, in K) at 2 m above sea level and analyzed the resulting maps for each pressure wave event to qualitatively determine where wave events occurred relative to surface air mass boundaries. $\theta_e$ was calculated following the approximation provided by (Bolton, 1980, their Eq. 43):

$$\theta_E = T_K(\frac{1000}{p})^{0.2854(1-0.28*10^{-3}r)} * exp[(\frac{3.376}{T_L} - 0.00254)*r(1+0.81*10^{-3}r)] \tag{7}$$

where $T_K$ is the temperature in K, $p$ is the air pressure in hPa, and $r$ is the water vapor mixing ratio (unitless). $T_L$ is the temperature at the lifting condensation level in K, approximated using Eq. 15 from Bolton (1980):

$$T_L = 56 + \frac{1}{\frac{1}{T_D - 56} + \frac{ln(T_K/T_D)}{800}} \tag{8}$$

where $T_D$ is the dew point temperature in K.

We analyzed maps of 300 hPa wind speed (in $m\,s^{-1}$) and geopotential height (in m) for each pressure wave event to determine where they occurred relative to upper-level troughs, ridges, and jet streaks. To quantify upper-level flow imbalance, we calculated the residual of the nonlinear imbalance equation on a constant 300 hPa surface ($\Delta NBE$ in $s^{-2}$; Zhang et al., 2000; Ruppert et al., 2022):

$$\Delta NBE = 2J(u,v) - \beta u + f\zeta - \nabla^2\phi \tag{9}$$





where $J(u, v)$ is the Jacobian of the horizontal flow (Eq. 10), $\beta$ ($\mathrm{s}^{-1}\,\mathrm{m}^{-1}$) is the change in the Coriolis parameter $f$ ($\mathrm{s}^{-1}$) with latitude, $u$ ($\mathrm{m\,s}^{-1}$) is the zonal component of the flow, $\zeta$ ($\mathrm{s}^{-1}$) is the vertical component of relative vorticity, and $\phi$ ($\mathrm{m}^2\,\mathrm{s}^{-2}$) is

the geopotential.

$$J(u, v) = \frac{\partial u}{\partial x}\frac{\partial v}{\partial y} - \frac{\partial u}{\partial y}\frac{\partial v}{\partial x} \tag{10}$$

Eq. 9 is obtained by scale analysis of the divergence tendency equation. Terms in the divergence tendency equation which contain the divergence, vertical velocity, and divergent components of horizontal velocity are dropped (Zhang et al., 2000). When the magnitude of $\Delta NBE$ is large relative to "background" (e.g., in straight, zonal flow) values, gravity wave generation

by balance adjustment may occur (James Ruppert, personal communication).

For each gravity wave event, we compared the location of the sensor network to the ERA5 mean sea level pressure (MSLP) patterns to determine where the detected waves occurred in a cyclone-relative framework. In some cases we were able to automatically track minima in ERA5 MSLP data (Tomkins et al., 2024a) using the algorithm described by Crawford et al. (2021). In the balance of cases we determined the cyclone location manually.

## 2.3  Radar data

We analyzed horizontal maps of reflectivity and radial velocity from the WSR-88D radars (NOAA National Weather Service Radar Operations Center, 1991) in Buffalo, NY (KBUF), for pressure wave events in Toronto, and Upton, NY (KOKX), for pressure wave events in New York and Long Island. An example of these maps for the event on 1 April 2023 is shown in Fig. 4. In each case, we used the scan at the $0.5°$ elevation angle.

Reflectivity bands were identified as roughly linear features of high reflectivity relative to the "background" reflectivity, following Tomkins et al. (2024b). To find features of locally-enhanced reflectivity, we first calculate the background reflectivity as a windowed average in radii of $20\,\mathrm{km}$. Grid points with reflectivity sufficiently exceeding that background average, or which have reflectivity $\geq 35\,\mathrm{dBZ}$, are identified as features (Fig. 4c). When mapping the reflectivity and detected high reflectivity features, we "mute" regions with enhanced reflectivity likely due to melting and mixed precipitation (reflectivity $> 20\,\mathrm{dBZ}$ and

correlation coefficient $< 0.97$) by plotting in greyscale (Fig. 4a,c; Tomkins et al., 2022). In Fig. 4, enhanced reflectivity features were found throughout the coverage of the KBUF radar, but most notably there were linear features near the eastern edge of Lake Erie and extending southeastward.

We identified Doppler velocity waves following Miller et al. (2022). We first calculate the difference in radial velocity between successive NWS WSR-88D scans. That difference field is then converted to a binary field; i.e., positive values are

converted to zeros, and negative values are converted to ones. Small objects are filtered out of the binary field. In Fig. 4d, Doppler velocity waves are detected across the radar domain, but most notably the wave extending from west-central Lake Ontario eastward then southward into New York State could be tracked as a coherent feature across several radar scans (Video supplement Animation-Figure-S3.01).



We analyzed the resulting sequences of maps for each wave event to determine whether any coherent bands or waves were present anywhere within the range of the radar. Additionally, if any bands or waves were present we assessed if they propagated directly over the pressure sensors at a velocity consistent with the estimated phase velocity of the pressure waves.

## 2.4 Surface stations and operational soundings

We used hourly ASOS data (NOAA National Centers for Environmental Information, 2021a) to assess precipitation type and liquid water equivalent precipitation amount during each pressure wave event. We counted the METAR snow precipitation type and snow mixed with other precipitation types as "snow". We tabulated the total number of hours during our analysis period in which at least a $0.1 \, \mathrm{mm/hr}$ of snowfall liquid water equivalent rate was measured. For New York, we were able to use data from John F. Kennedy International Airport (KJFK) to obtain both precipitation type and snowfall intensity. As precipitation amounts at Toronto Pearson International Airport (CYYZ) were not available in the archived data, we used precipitation amount data from the Downtown Toronto (CXTO) ASOS, which does not record precipitation type, and precipitation type data from the Toronto City Airport (CYTZ) ASOS, which is closer to CXTO than CYYZ but does not record precipitation amount.

We used one-minute ASOS data to help determine whether each wave event was directly caused by the passage of a front or outflow boundary and its associated density and temperature change (e.g., when a sharp rise in pressure co-occurred with sharp drops in temperature and dew point). We also considered the radar and surface analysis maps when determining whether an event was directly caused by a front or outflow boundary passage. Examples of pressure wave events associated with an outflow boundary passage and a frontal passage, along with contextual data, are shown by Allen et al. (2024d), their Sects. 4.3 and 4.4, respectively. In one case, a pressure wave event was caused by a convective wake low passage as indicated by the timing of the event relative to a mesoscale convective system passage (Allen et al., 2024d). Ideally, if we saw surface wind perturbations correlated with the pressure perturbations in one-minute ASOS data, then that would strongly suggest the pressure perturbations are associated with gravity waves. But, $< 1\text{-min}$ surface wind data sets at the locations of the pressure sensor network sensors are not available. ERA5 and other reanalysis are too coarse in spatial and temporal scale to use for this purpose. We separated the front, outflow boundary, and wake low cases from the remaining cases, which we refer to as gravity wave events.

We analyzed upper-air radiosonde observations for gravity wave events with a nearby NWS weather balloon (Fig. 2) launched during a time window from 2 hours before the start of the wave event to 2 hours after the end of the wave event. For gravity wave events in the Toronto pressure sensor network, radiosonde data from Buffalo, NY (KBUF), were used, and for gravity wave events in the New York and Long Island pressure sensor network, radiosonde data from Upton, NY (KOKX), were used. We obtained the data from IGRA (NOAA National Centers for Environmental Information, 2021b) and interpolated the data to a constant 100-meter vertical resolution. When sounding data were available, we determined whether an efficient wave duct was present (conditionally unstable layer above an absolutely stable layer; Lindzen and Tung, 1976), as in Allen et al. (2024d), their Fig. 10, which added confidence that the pressure wave event was associated with gravity waves. For each sounding associated with a gravity wave event, we will determine whether a surface-adjacent temperature inversion was present



or if any temperature inversions were present in the lowest 1 km above the surface. We identified temperature inversion layers as any observations where the temperature increased with increasing height.

## 3 Pressure wave characteristics and environmental context

Table 3 lists important attributes of all 33 pressure wave events, and it labels the events with numbers which will be referred to in the following text. No pressure wave events were detected between June and August (Fig. 5a). Five pressure wave events were solitary waves coincident with frontal passages, 4 pressure wave events were coincident with outflow boundary passages, 1 pressure wave event was caused by a wake low associated with a mesoscale convective system (Allen et al., 2024d), and the other 23 pressure wave events are considered gravity wave events (Fig. 5b,c).

There did not appear to be a strong relationship between wave period and wave amplitude for pressure wave events (Fig. 5b), which is somewhat surprising, given that the mean wavelet power generally increases with wave period for pressure (Canavero and Einaudi, 1987; Grivet-Talocia and Einaudi, 1998; Allen et al., 2024d). Individual pressure wave events (Fig. 5) may not follow the same pattern of increasing amplitude with increasing wave period as seen in longer-term mean values of wavelet power (Allen et al., 2024d, their Fig. 5). The pressure wave events are caused by atypical short-term pressure perturbations

whereas the long-term mean wavelet power mainly consists of quiescent conditions, usually without sharp pressure changes. Figure 5b includes the range of wave periods where the wavelet power exceeded $A(a)$ as error bars. From these error bars, it is apparent that nearly every pressure wave event had a strong wave signal at shorter wave periods (< 30 min), while very few had a strong wave signal at longer wave periods (> 90 min).

Every pressure wave event had an eastward component to its phase velocity (Fig. 5c). This result is similar to Grivet-

295 Talocia et al. (1999), who found that 95% of pressure wave events in central Illinois had an eastward component to their phase velocities. Nineteen of the pressure wave events (58%) we detected had a northward component to their phase velocities, while 14 pressure wave events (42%) had a southward component to their phase velocities. Twenty out of 33 (61%) pressure wave events we detected had a phase speed between 20 m s$^{-1}$ and 35 m s$^{-1}$, again similar to Grivet-Talocia et al. (1999).

### 3.1 Gravity wave event characteristics

We will focus on the 23 gravity wave events to address their environmental and radar contexts with a focus on winter storms. All 23 gravity wave events occurred between December and May (Table 3). Figure 6 shows the extracted event and total pressure time series for a single sensor for each of those 23 gravity wave events. Most events consisted of multiple pressure oscillations. In some cases the amplitudes of those oscillations varied with time (e.g., events 2, 14, and 24), while in others the oscillations remained at a steady amplitude through the event (e.g., events 11 and 18). The gravity wave events had a wide

range of durations, wave amplitudes, and wavelengths (Fig. 5). Event duration varied over a wide range. Event 5 was a solitary wave of depression with a duration of roughly 1 hour. Event 22 had a duration of nearly 20 hours.

For the 23 gravity wave events, a strong linear correlation between wave amplitude and event duration was found (R = 0.88, p-value: $3.2 \times 10^{-8}$, Fig. 7). A simple linear regression suggests that a 1 hPa increase in amplitude corresponds roughly to a





170 min increase in event duration. It is possible that part of this correlation is due to the event extraction method. Testing on
synthetic events with constant duration (not shown) showed that the higher-amplitude waves result in more residual wavelet
signal extending beyond the given event duration. Given the large range of event durations over which this correlation holds,
there is likely some physical meaning to the relationship. A similar relationship has been documented in seismic waves: higher-
magnitude earthquakes tend to have longer durations (e.g., Trifunac and Brady, 1975; Herrmann, 1975), which can be explained
by the stronger earthquakes propagating over larger areas of fault surfaces (e.g. Bonilla et al., 1984; Wells and Coppersmith,
1994) and thus having larger source areas. This raises the question of whether higher-amplitude gravity waves have larger
source areas, which is not possible to adequately answer with the data used in this study.

### 3.1.1    Relating pressure perturbations to vertical parcel displacements for gravity waves

To give further context to the pressure perturbations associated with gravity waves, we can compute the vertical parcel per-
turbation for a case in which representative sounding data are available. The sounding launched at KBUF during Event 10 in
Toronto on 25 February 2022 is useful for this as there is a clear gravity wave ducting layer in that example (Allen et al., 2024d,
their Fig. 10). Equation 68-3 from Gossard and Hooke (1975) relates the pressure perturbation ($P_0$ in Pa) to the vertical parcel
displacement ($\zeta_H$ in m) for a given gravity wave ducting layer depth (H in m):

$$\zeta_H = \frac{H}{\rho_s (\frac{\omega}{k})^2} B P_0 \tag{11}$$

where $\rho_s$ is the surface air density ($1.225 \ \mathrm{kg \ m^{-3}}$ for this example), $\omega$ is the intrinsic angular wave frequency (in $\mathrm{s^{-1}}$), and k
is the horizontal wavenumber (in $\mathrm{m^{-1}}$). $\omega$ and k are calculated by:

$$\omega = \frac{2\pi}{\tau} - u_0 k \tag{12}$$

$$k = \frac{2\pi}{\lambda} \tag{13}$$

where $\tau$ is the wave period (1222 s for this example; Table 3), $u_0$ is the mean wind speed within the wave duct ($18.9 \ \mathrm{m \ s^{-1}}$ for
this example), and $\lambda$ is the wavelength (55.5 km for this example; Table 3). For this example, $\omega = 0.003 \ \mathrm{s^{-1}}$ and $k = 0.11 \ \mathrm{km^{-1}}$.
The calculation of B depends on the Brunt-Väisälä frequency (N in $\mathrm{s^{-1}}$). The wave duct is saturated for this example (Allen
et al., 2024d, their Fig. 10), so we calculate the moist Brunt-Väisälä frequency $N_m$ (Markowski and Richardson, 2010, p. 42):

$$N_m = \sqrt{\frac{g}{\theta_{e0}} \frac{\Gamma_m}{\Gamma_d} \frac{\partial \theta_e}{\partial z}} \tag{14}$$

where g is the gravitational acceleration ($\sim 9.81 \ \mathrm{m \ s^{-2}}$), $\theta_{e0}$ is the mean equivalent potential temperature in the wave duct
(294.7 K for this example), $\Gamma_m$ is the moist adiabatic lapse rate in the wave duct ($7.58 \ \mathrm{K \ km^{-1}}$ for this example), $\Gamma_d$ is the
dry adiabatic lapse rate ($9.76 \ \mathrm{K \ km^{-1}}$), and $\frac{\partial \theta_e}{\partial z}$ is the change in equivalent potential temperature with height in the wave duct
($20.2 \ \mathrm{K \ km^{-1}}$ for this example). For this example, $N_m = 0.023 \ \mathrm{s^{-1}}$. We also need the vertical wavenumber $n_1$ (in $\mathrm{m^{-1}}$) to



calculate B. $n_1$ is calculated by:

$$n_1 = k\sqrt{\frac{N_m{}^2}{\omega} - 1} \tag{15}$$

For this example, $n_1 = 0.853$ km$^{-1}$. Since $\omega < N_m$, B is calculated by (Gossard and Hooke, 1975):

$$B = \frac{sin(n_1 H)}{n_1 H} \tag{16}$$

For this example, B = 0.656. Finally, the peak-to-trough amplitude of the gravity wave was $\sim$2 hPa for Event 10 (Table 3), so we take half of that, 1 hPa, as the pressure perturbation $P_0$, so the vertical parcel displacement $\zeta_H = 129$ m for this example. This result is on a similar order of magnitude to the vertical displacements reported by (Kjelaas et al., 1974, 50-120 m) and

345 (Allen et al., 2013, 400 m).

### 3.1.2 Synoptic context for gravity wave events

The synoptic environment setting for each of the 23 gravity wave events that occurred during our 40 months of analysis (Table 4) puts these events in context and permits comparisons to previous case studies and theoretical work. For each gravity wave event, we examined surface pressure and equivalent potential temperature (Fig. 8), 300 hPa geopotential heights and

350 wind speeds (Fig. 9), and 300 hPa $\Delta NBE$ (Fig. 10). For a gravity wave to be detected at the surface, there needs to be suitable conditions for the wave signal to reach the surface (e.g., there should ideally be no convective overturning in the boundary layer which would obscure the pressure signal due to the gravity wave).

Surface low center cyclone tracks for storms which produce snowfall in the Northeast United States are most common near the coast and over the Atlantic Ocean, to the east of New York City and Toronto (Fig. 11). As regards suitable near surface

conditions for gravity waves, of the 23 gravity wave events in Toronto and New York during our 40-month analysis period, 13 (57%) occurred north or east of a surface low (events 3, 4, 10, 11, 12, 13, 16, 17, 18, 19, 22, 24, and 28), often on the cool side of warm or stationary fronts. Event 22 had such a long duration that it began when Toronto was on the cool side of a warm front and ended after the warm front had passed. Event 25 and 30 occurred very far (2000 km or more) to the east of a surface low and on the cool side of an air mass boundary. Event 2 occurred behind a cold front and to the west of a surface low. Event

6 occurred near a weak surface low and just on the cool side of an air mass boundary. Events 5, 9, and 27 occurred south of lows. Event 26 occurred near a weak air mass boundary with lows both to the north and the south.

Inversion layers at altitudes < 1 km were found in all of the 12 gravity wave events when upper air soundings were launched either during or within 2 hours of the events (Figure 12). Event 22 had two radiosonde launches. Many of the inversion layers were only 100-200 m deep. Events 6, 17, 22, and 25 had an inversion layer adjacent to the surface. A near surface stable layer

likely helps to maintain the coherence of the gravity wave signal across the network of sensors (Uccellini and Koch, 1987). Unfortunately, coincident upper air soundings were not available for events 7 and 14 when gravity waves occurred with no closed surface low anywhere in the domain we analyzed (Fig. 8).

In terms of the large scale synoptic pattern aloft, 13 gravity wave events occurred downstream of 300 hPa troughs and upstream of 300 hPa ridges (Fig. 9, Table 4, events 2, 3, 4, 5, 10, 12, 14, 16, 22, 24, 26, 27, and 28), consistent with most



gravity wave events shown by Uccellini and Koch (1987). Six others occurred in roughly zonal 300 hPa flow regimes (events
9, 11, 17, 18, 25, and 30), and 3 gravity wave events occurred below a 300 hPa ridge (events 6, 13, 19). One gravity wave
event occurred upstream of a 300 hPa trough (event 7).

Regions with large magnitude $\Delta NBE$, regardless of sign, imply flow imbalance and the possibility of resulting gravity wave
genesis. If gravity waves are triggered by flow imbalance at 300 hPa, they would not necessarily be observed on the ground
directly beneath the trigger area as the wave signal must reach the lower troposphere to be observed, which might require
the waves to propagate some distance vertically and horizontally. Eighteen of the 23 gravity wave events occurred with large
300 hPa flow imbalance to the south or west (events 2, 3, 5, 6, 7, 9, 10, 11, 13, 14, 16, 18, 19, 22, 24, 25, 27, and 30) (Fig. 10).
Considering that many of the gravity wave events were observed to propagate from west to east (Fig. 5c and Table 3), it is
plausible that many were triggered by flow imbalance aloft.

In general, previous studies (e.g., Uccellini and Koch, 1987; Koch and Dorian, 1988) often found mesoscale gravity waves
east of surface lows and downstream of upper-level troughs. Our analysis of the 23 gravity wave events in the Toronto and New
York metro areas between January 2020 and April 2023 largely agrees with those findings. In such cases, gravity waves may
have been triggered by the balance adjustment mechanism described by Zhang et al. (2001). The gravity wave events associated
with 300 hPa zonal flow with weak or no flow imbalance in the region were likely associated with different mechanisms, such
as localized latent heating or interactions between waves propagating from farther afield (Fritts and Alexander, 2003). However,
with the available observations and reanalysis data, it is not possible to determine the gravity wave trigger mechanism with
complete certainty.

### 3.1.3   Radar echo and precipitation type context for gravity waves

NWS WSR-88D radar echo corresponds to precipitation-sized particles in the resolution volume. Only within regions with
radar echo can enhanced reflectivity features and Doppler velocity waves be detected. Table 5 shows the radar echo character-
istics and ASOS precipitation type context for each gravity wave event.

Eighteen (78%) of the 23 gravity wave events occurred with precipitation radar echo detected by the nearby WSR-88D in
the 0.5° tilt. Only 6 of these cases (events 2, 4, 10, 12, 18, and 28) co-occurred with surface snow or mixtures including snow.
Two of those cases with snow occurred with enhanced reflectivity bands within the radar range, but in neither case was the
movement of the enhanced reflectivity bands consistent with the gravity wave phase velocity vector. For example, radar data
during event 28 indicate there was an enhanced reflectivity feature passing over the pressure sensors, but the movement of the
enhanced reflectivity feature (SW to NE) was not consistent with the phase direction of the gravity waves (NW to SE) (Fig. 13
and Video supplement Animation-Figure-S3.02).

Overall, gravity waves during surface snow were rare at our locations. Periods of snowfall at a rate of at least $0.1\ \mathrm{mm\,hr^{-1}}$
(liquid equivalent) for at least 4 hours, with at most a one-hour gap without that rate of snowfall, occurred 59 times in Toronto
and 20 times in New York during our analysis period. Fifty-one of those 59 snow storms in Toronto, and 16 of the 20 in New
York, occurred between November and February, mostly before the peak in gravity wave events (February-May; Fig. 5a). In
the Toronto area, there were 460 hours with at least $0.1\ \mathrm{mm\,hr^{-1}}$ (liquid-equivalent) of snow recorded. Of those, only 15 hours



with snow were during a gravity wave event. In the New York area, snow was recorded for 134 hours of which only 4 occurred
during gravity wave events (Table 6).

When surface rain was present, 3 gravity wave cases (events 22, 24, and 26) had enhanced reflectivity features collocated
and moving at a velocity consistent with the pressure waves. During event 22, an elongated reflectivity feature crossed the
pressure sensor network and appeared to move at a velocity consistent with the gravity wave phase velocity (Fig. 14, and
Video supplement Animation-Figure-S3.03). The reflectivity band was an isolated feature lasting only 2 hours in a wave event
which lasted nearly 20 hours. Event 24 occurred along with an occluded front which passed over the pressure sensor network
at 20 UTC on 25 Jan 2020. We chose categorize event 24 as a gravity wave rather than a "front" event because of the pressure
oscillations observed in the hours before the occluded front passage (Fig. 6). During event 26, a narrow region of enhanced
reflectivity on the trailing edge of a broader precipitation region passed over the pressure sensors in New York near the same
time as a large pressure minimum (6:45 UTC on 1 May 2020; Fig. 6).

Depending on the spatial scale of gravity waves and the height and depth of the wave duct as well as their 3D position
relative to the slanting WSR-88D scans, the transient convergence and divergence signals associated with the gravity wave's
propagating upward and downward motions may or may not yield radar detectable Doppler velocity waves. Hence, we do not
expect a 1:1 correspondence between detected gravity waves and detected Doppler velocity waves in the 0.5° elevation angle
scan.

Thirteen gravity wave events of the 18 gravity waves with radar echo occurred with coherently moving Doppler velocity
waves present anywhere within the range of the nearby WSR-88D radar. But, only five of these had Doppler velocity waves
over the pressure sensors moving at a velocity consistent with the gravity wave phase velocity (Table 5). Based on this limited
evidence, a subset of gravity waves may manifest a Doppler velocity wave signature. Figure 3 and Eq. 2 from Allen et al.
(2024d) suggest that any gravity wave which produces a pressure perturbation $\geq 0.5$ hPa should also produce a detectable
velocity wave signal. The velocity waves may not appear in radar Doppler velocity data either because they are above or below
the height of the radar beam, or because there is strong turbulence which obscures the signal associated with gravity waves. In
Toronto, the KBUF radar beam is at a higher altitude than the KOKX radar beam is at over New York, and 3 of the 5 gravity
wave events with corresponding Doppler velocity waves occurred in the New York pressure sensor network.

The Doppler velocity wave detection works best for waves which propagate less than half of their wavelength between
successive radar scans (Miller et al., 2022), i.e., which have a wave period at least twice as long as the time between successive
radar scans. Typical NEXRAD volume coverage patterns have a ∼4-8 minute time between 0.5° elevation scans, and 12 of
the 18 gravity wave events which co-occurred with radar echo had a wave period of 16:39 or less. Only 2 of those 12 gravity
wave events were collocated with Doppler velocity waves which propagated at a velocity consistent with the gravity wave
phase velocity. Of the 6 gravity wave events with a wave period longer than 16:39 that co-occurred with radar echo, 3 were
collocated with Doppler velocity waves which propagated at a velocity consistent with the gravity wave phase velocity.



## 4 Conclusions

We deployed two air pressure sensor networks, one in Toronto, ON, Canada, and the other in the New York City area and Long Island, NY, USA, to study atmospheric gravity waves. In over 3 years of data, we objectively identified 33 pressure wave events which were observed by at least 4 pressure sensors and for which there was reasonable confidence in the estimate of

440 the wave phase velocities. Our study examined wave amplitudes on the order of 0.5-5 hPa and wave periods on the order of 2-67 min. These spatial and temporal scales were chosen to align with the spatial and temporal scales of radar-observed enhanced reflectivity bands and Doppler velocity waves, both of which were surmised to potentially be related to gravity waves (Hoban, 2016; Miller et al., 2022).

A few of our detected pressure wave events were associated with frontal passages (5), outflows (4), and a wake low (1), and

445 the remaining 23 were gravity waves, 20 of which occurred in the cool season between November and April. For context, there were 20 snow storms in the New York and 59 in the Toronto metropolitan areas over our 40 month observation period. While limited, the evidence we have suggests a both a lack of a common associations between reflectivity bands and gravity waves and between Doppler velocity waves and gravity waves. Just 6 of the gravity wave events co-occurred with any surface snowfall (including trace amounts). Only 2 of those 6 events had any enhanced reflectivity bands in the vicinity. The spatial wavelengths

of the gravity waves and enhanced reflectivity bands were similar, but in all the cases with snow, the reflectivity bands were either not directly over the pressure sensors or not moving at a velocity consistent with the pressure waves (Table 5). A subset of our detectable gravity waves (5 of 18 gravity waves with radar echo in vicinity) may have manifested a detectable Doppler velocity wave signature in NEXRAD data but most did not.

While the observational and reanalysis output data used cannot confirm the cause of gravity wave genesis, most gravity wave

events occurred with strong upper-level flow imbalance to the south or west of their location, suggesting that the mechanism of balance adjustment described by Zhang (2004) and Ruppert et al. (2022) may be relevant. The occurrence of several gravity waves downstream of an upper level trough, on the cool side of air mass boundaries, and with a temperature inversion in the lowest 1 km above ground level is consistent with the findings of Uccellini and Koch (1987).

We found a strong linear relationship between amplitude and event duration for the 23 atmospheric gravity wave events

detected (Fig. 7). A potential explanation could be that parcels in gravity waves triggered by a larger initial perturbation might oscillate for a longer time before returning to an equilibrium state. Further exploration of the relationship between gravity wave amplitude and duration is a topic for future research. There may be an analogy to seismic waves in that higher-amplitude earthquakes tend to have longer durations because of the larger rupture area along the fault (Trifunac and Brady, 1975; Herrmann, 1975).

Satellite images of northeast US winter storms often show undulations in the overlying cirrus. These undulations may be either Kelvin-Helmholtz waves or gravity waves. Kelvin-Helmholtz waves on horizontal scales of ∼3 km could locally alter the cloud microphysical properties (Houser and Bluestein, 2011). The surface pressure should reflect changes throughout the column of air, including gravity waves aloft. However, it is possible that gravity waves in the upper cloud layers with periods between 3 and 67 min do occur but have their surface pressure signals obfuscated by other perturbations. For example, if there





is an unstable layer below the layer with the gravity waves, then the pressure wave amplitude at the surface would be reduced
and obfuscated by pressure perturbations due to convective overturning (Kjelaas et al., 1974). Wind profiler data would help to
resolve whether conditions for Kelvin-Helmholtz waves are present within the cirrus layer.

For the New York City area in particular, the low frequency of occurrence of gravity waves in winter storms is influenced
in part by a sample bias related to the typical position of low pressure centers off-shore. The New York City metropolitan area
and Long Island are usually in the northwest quadrant of the storm where gravity waves are not often found (Fig. 11). In both
Toronto and New York, most snow storms $\geq 4$ h in duration occurred between December and February, while most gravity
waves were detected between February and May (Fig. 5a).

Whereas previous case study work examined heavy snow events that had gravity waves, we cast a broad net by putting out
pressure sensor networks for an extended time period to see what we could "catch". Some of the previously studied winter storm
gravity wave cases (e.g., Bosart et al., 1998) are clearly not representative of typical flat-land winter storms in the Northeast
US, since of the 79 winter storms with snow that occurred over our 40 months of observations, only 6 had detectable gravity
waves. It is well established that gravity waves can locally increase precipitation (e.g., Bosart et al., 1998; Gaffin et al., 2003;
Colle, 2004; Allen et al., 2013; Kingsmill et al., 2016). But, if gravity waves of a sufficient amplitude do not occur then they
are irrelevant to locally increasing snow rates. Our findings suggest that gravity waves of amplitude $\geq 0.5$ hPa are much less
common in winter storms than reflectivity features on similar spatial and temporal scales, which are present in most winter
storms (Hoban, 2016; Ganetis et al., 2018).

*Code and data availability.* Data: The specific data shown in each figure can be found at https:doi.org/10.5281/zenodo.11286349 (Allen,
2024). The pressure time series data used throughout this publication can be found at https://doi.org/10.5281/zenodo.8136536 (Miller and
Allen, 2023). The NWS NEXRAD Level-II data used in Figs. 4, 13 and 14 can be accessed from the National Centers for Environmental
Information (NCEI) at https://www.ncei.noaa.gov/products/radar/next-generation-weather-radar (NOAA National Weather Service Radar
Operations Center, 1991). The one-minute ASOS data can be accessed from NCEI at https://www.ncei.noaa.gov/products/land-based-station/
automated-surface-weather-observing-systems (NOAA National Centers for Environmental Information, 2021a), and hourly ASOS data
can be accessed from NCEP at https://madis-data.ncep.noaa.gov/. The radiosonde data used to create Fig. 12 can be accessed from NCEI
at https://www.ncei.noaa.gov/products/weather-balloon/integrated-global-radiosonde-archive (NOAA National Centers for Environmental
Information, 2021b).

Code: The code used for processing the pressure time series data can be found at https://doi.org/10.5281/zenodo.8087843 (Allen and
Miller, 2023).

*Video supplement.* List of animations with captions and filenames

All animations can be viewed at: https://av.tib.eu/series/1721/video+supplement+to+objectively+identified+mesoscale+surface+
air+pressure+waves+in+the+context+of+winter+storm+environments+and+radar+reflectivity+features+a+3+year+analysis. In-
dividual animations can be viewed by following the DOI URL.



Animation-Figure-S01: Animated maps of **(a)** reflectivity, **(b)** Doppler velocity, **(c)** enhanced reflectivity feature detection and **(d)** Doppler velocity wave detection for NWS WSR-88D radar data from Buffalo, NY, at 0.5° tilt, from 04:00 UTC to 11:05 UTC on 1 April 2023. In **(a)** and **(c)**, values are shown in greyscale when there is likely enhancement due to melting (Tomkins et al., 2022). Filled blue circles indicate locations of pressure sensors which captured pressure wave event 15, and unfilled blue circles indicate locations of pressure sensors which did not capture the pressure wave event. This animation goes with Fig. 4. Title: 2023/04/01 KBUF radar 4-panel animation. https://doi.org/10.5446/67635 (Allen et al., 2024b).

Animation-Figure-S02: Animated maps of **(a)** reflectivity, **(b)** Doppler velocity, **(c)** enhanced reflectivity feature detection and **(d)** Doppler velocity wave detection for NWS WSR-88D radar data from Upton, NY, at 0.5° tilt, from 15:59 UTC to 22:26 UTC on 18 February 2021. In **(a)** and **(c)**, values are shown in greyscale when there is likely enhancement due to melting (Tomkins et al., 2022). Filled blue circles indicate locations of pressure sensors which captured pressure wave event 26, and unfilled blue circles indicate locations of pressure sensors which did not capture the pressure wave event. This animation goes with Fig. 13. Title: 2021/02/18 KOKX radar 4-panel animation. https://doi.org/10.5446/67765 (Allen et al., 2024a).

Animation-Figure-S03: Animated maps of **(a)** reflectivity, **(b)** Doppler velocity, **(c)** enhanced reflectivity feature detection and **(d)** Doppler velocity wave detection for NWS WSR-88D radar data from Upton, NY, at 0.5° tilt, from 07:49 UTC on 5 April 2023 to 04:00 UTC on 6 April 2023. In **(a)** and **(c)**, values are shown in greyscale when there is likely enhancement due to melting (Tomkins et al., 2022). Filled blue circles indicate locations of pressure sensors which captured pressure wave event 18, and unfilled blue circles indicate locations of pressure sensors which did not capture the pressure wave event. This animation goes with Fig. 14. Title: 2023/04/05 KBUF radar 4-panel animation. https://doi.org/10.5446/67633 (Allen et al., 2024c).

*Author contributions.* LRA, SEY, and MAM conceptualized the project and designed the methodology. MAM designed and built the pressure sensors and managed the pressure sensor networks. LRA and MAM wrote the data processing software. LRA and LMT created the visualizations with input from SEY and MAM. LRA prepared the manuscript, SEY edited the manuscript, and MAM and LMT contributed to the final stages of reviewing and editing.

*Competing interests.* The contact author has declared that none of the authors have any competing interests.

*Acknowledgements.* The authors express their sincere appreciation to the pressure sensor hosts in Toronto and the New York City metro area and Long Island, including colleagues from Environment Canada, Stony Brook University, Columbia University, and friends, including Jase Bernhardt, Drew Claybrook, Brian Colle, Daniel Horn, Daniel Michelson, Robert Pincus, Adam Sobel, David Stark, and Jeff Waldstreicher, who graciously agreed to plug in pressure sensors to their home internet. The development of the methodology, interpretation of the results, and visualizations benefited from discussions and correspondence with DelWayne Bohnenstiehl, Brian Colle, Declan Crowe, Brian Mapes, Logan McLaurin, Sonia Lasher-Trapp, Matthew Parker, James Ruppert, and Minghua Zhang.





This work was supported by the National Science Foundation (AGS-1905736), the National Aeronautics and Space Administration (80NSSC19K0354), the Office of Naval Research (N000142112116 and N000142412216), and the Center for Geospatial Analytics at North Carolina State University.



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





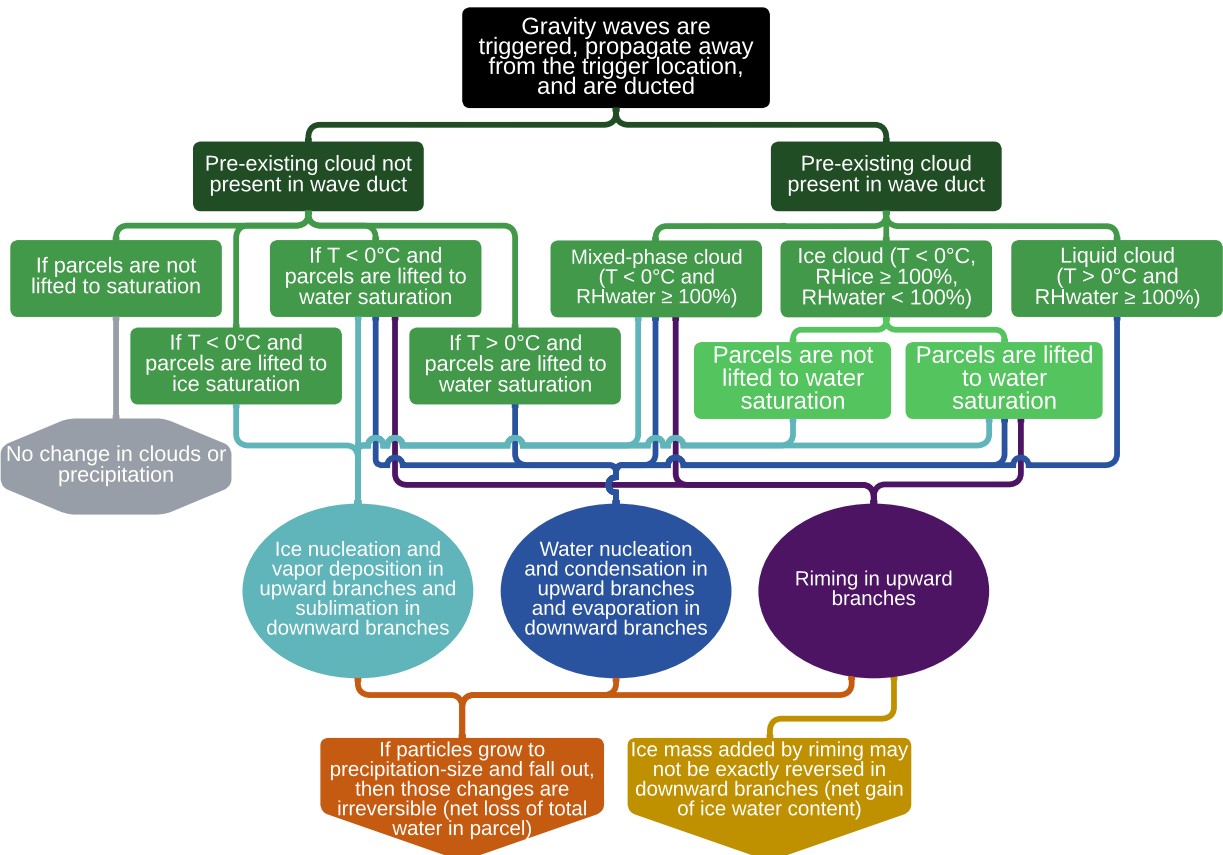

**Figure 1.** Possible chains of processes and outcomes for gravity waves to yield changes in cloudiness and precipitation. The sequence goes from top to bottom. Green rectangles indicate conditions or requirements, ovals indicate microphysical processes which result from the air motions associated with gravity waves, and downward-pointing pentagons indicate irreversible changes within air parcels resulting from the previous steps. For simplicity, sequences where air parcel temperatures cross the 0°C level altitude are not shown.





**Figure 2.** Maps of the pressure sensor, ASOS, and radar sites used in this study. **(a)** Locations of Toronto and New York City. **(b)** Pressure sensor sites in Toronto (filled circles) with the 3 ASOS sites (stars) and KBUF radar (maroon diamond). **(c)** Pressure sensor sites in New York and Long Island (filled circles) with KJFK ASOS (black star) and KOKX radar (maroon diamond).





**Figure 3.** Timelines of when the pressure sensors in **(a)** Toronto and **(b)** New York and Long Island recorded pressure data between January 2020 and April 2023.





**Figure 4.** Event 19 (gravity wave event) maps of **(a)** reflectivity, **(b)** radial Doppler velocity, **(c)** detected features of enhanced reflectivity, and **(d)** Doppler velocity wave detection for KBUF at 05:54 UTC on 1 April 2023. Filled blue circles indicate the locations of pressure sensors in Toronto. In this example, gravity waves moved SW to NE while a NW-SE aligned linear region of enhanced reflectivity about 150 km long and 80 km wide extends from near the western edge of Lake Ontario. Several NW-SE aligned Doppler velocity waves could be tracked from SW to NE between Lake Erie and Lake Ontario. The greyscale regions in **(a)** and **(c)** likely contain mixed precipitation (reflectivity > 20 dBZ and dual-polarization correlation coefficient < 0.97). An animated version of this figure is available in the Video Supplement Animation-Figure-S01.







**Figure 5.** Characteristics of the 33 pressure wave events detected in New York (orange) and Toronto (green) between January 2020 and April 2023. **(a)** Bar chart of the number of pressure wave events by month. **(b)** Scatter plot of wave amplitude against wave period, with error bars indicating the range of wave periods where $W \geq 10 \langle |W(b,a)| \rangle_b$, i.e., where there was a strong wave signal. **(c)** Radial scatter plot of the wave phase velocities (directions shown are in degrees clockwise from northbound). In panels **(b)** and **(c)**, gravity wave events are indicated by filled circles, and front, outflow, and wake low events are indicated by other shapes according to the legend.





**Figure 6.** Extracted pressure wave event (black, left axes) and total pressure (blue, right axes) time series for the 23 gravity wave events. The ordering and numbering of wave events matches that in Table 3. For each gravity wave event, data from only a single sensor are shown. That sensor was chosen to maximize its optimal cross-correlation values with extracted event traces from other sensors which captured a given gravity wave event.







**Figure 7.** Scatter plot of wave amplitude against event duration for the 23 gravity wave events. Green points represent gravity wave events in Toronto, and orange points represent events in New York and Long Island.





## 2-meter $\theta_e$ and MSLP for gravity wave events



**Figure 8.** ERA5 2 m equivalent potential temperature maps for all of the detected gravity wave events, at the center time of each event. The ordering and numbering of events matches that in Table 3. MSLP is contoured in white every 5 hPa. In each panel, either New York or Toronto are shown by cyan points, depending on where the gravity wave event occurred.







**Figure 9.** ERA5 300 hPa wind speed maps for all of the detected gravity wave events, at the center time of each event. The ordering and numbering of events matches that in Table 3. 300 hPa geopotential height is contoured in black every 50 m. In each panel, either New York or Toronto are shown by cyan points, depending on where the gravity wave event occurred.





**Figure 10.** ERA5 300 hPa ΔNBE maps for all of the detected gravity wave events, at the center time of each event. The ordering and numbering of events matches that in Table 3. In each panel, either New York or Toronto are shown by magenta points, depending on where the gravity wave event occurred.





**Figure 11.** Cyclone track density (shading) for storms in the Northeast United States (NEUS) which brought at least 1 in of snowfall in a 24 hour period to at least two ASOS stations in the NEUS between 1996 and 2023. Cyclones were tracked using ERA5 data following the methodology of Crawford et al. (2021).



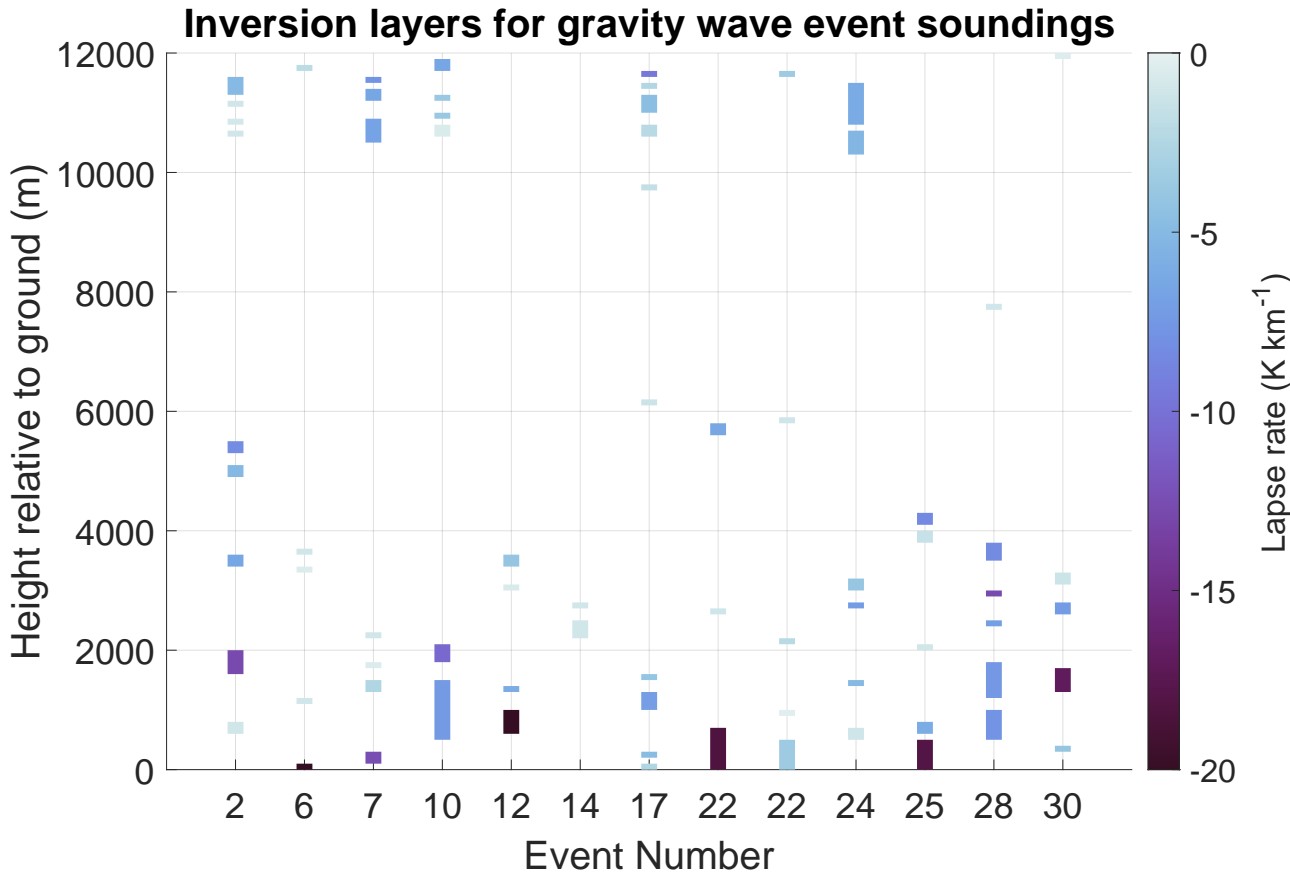

**Figure 12.** Air temperature inversion layers for soundings launched at KBUF for gravity wave events in Toronto, or at KOKX for gravity wave events in New York and Long Island, either during or within 2 hours of a gravity wave event. Inversion layers are colored according to the layer average lapse rate (darker colors indicate a stronger inversion). Event numbering matches that in Table 3. Two soundings were launched at KBUF during event 22. Events 2-22 were in Toronto and 24-30 were in NY.





**Figure 13.** As in Fig. 4, but for KOKX at 18:30 UTC on 18 February 2021, during event 28 (gravity wave event). In this example, an elongated enhanced reflectivity feature passed over the pressure sensor network from SW to NE during the wave event, which was inconsistent with the gravity wave phase direction (NW to SE). An animated version of this figure is available in the Video Supplement Animation-Figure-S02.





**Figure 14.** As in Fig. 4, but for KBUF at 19:30 UTC on 5 April 2023 during event 22 (gravity wave event). In this example, gravity waves moved from NW to SE while an elongated enhanced reflectivity feature is passing over the pressure sensors in Toronto from west to east. An animated version of this figure is available in the Video Supplement Animation-Figure-S03.



| Term | Conventional symbol | Definition |
|---|---|---|
| Wave period | $\tau$ | For a fixed location, the amount of time from one wave peak (or trough) to the next. Inverse of wave frequency. Unit: s. |
| Wave frequency | $f$ | For a fixed location, the number of wave peak (or trough) passages per unit time. Inverse of wave period. Units: $\mathrm{s}^{-1}$ or Hz. |
| Wavelength | $\lambda$ | The distance between wave peaks (or wave troughs). Product of wave period and phase speed. Unit: m. |
| Wave amplitude | A | The difference between the wave peak and trough values in hPa. |
| Event duration | | The amount of time over which a detectable wave signal was present. |
| Wavelet power | $|W|$ | The absolute value of the wavelet transform output. Units: $\mathrm{hPa}^2\,\mathrm{s}^{-1}$ (when applied to pressure in hPa). |
| Phase speed | $|\boldsymbol{c}|$ or $c$ | The distance traversed by a wave peak (or wave trough) per unit time. Product of wavelength and wave frequency. Units: $\mathrm{m\,s}^{-1}$. |
| Phase direction | | The direction relative to north in which the wave peaks/troughs are propagating. Unit: °. |
| Phase velocity | $\boldsymbol{c}$ | Vector in xy-plane which has magnitude defined by the phase speed and direction defined by the phase direction. Units: $\mathrm{m\,s}^{-1}$. |
| Slowness vector | $\boldsymbol{s}$ | Vector in xy-plane with components equal to the inverse of the phase velocity components. Points in phase direction. Units: $\mathrm{s\,m}^{-1}$. |
| Pressure wave | | Detectable wave signals present in time series of pressure over at least 4 pressure sensors in a network. |
| Gravity wave / buoyancy wave | | Waves attributed to the physical mechanism by which air parcels in a stable environment, when perturbed vertically, will oscillate about their original altitude. |
| Doppler velocity wave | | Sets of banded features in radar Doppler velocity data. Detected from scanning radar data following Miller et al. (2022). |

**Table 1.** Definitions of key terms used in this paper.



| Paper | Detection method | Number of cases | Amplitudes | Wavelengths | Wave periods | Phase speeds | Event durations | Types of waves, geographic region |
|---|---|---|---|---|---|---|---|---|
| Christie et al. (1978) | Pressure sensors | 99 | 0.05-1.1 hPa | 0.4-30 km | *2-20 min | 4-50 m s$^{-1}$ | *8 min - 2 h | Wave packets and solitary waves; Central Australia |
| Kjelaas et al. (1974) | Acoustic sounders / pressure sensors | 3 | 50-120 m vertical displacement | 1-6 km | 3-12 min | 5-10 m s$^{-1}$ | 20-40 min | Wave packets; Haswell, Colorado |
| **This study** | **Pressure sensors** | **33** | **0.45-5.51 hPa** | **3.5-170 km** | **2-67 min** | **17-62 m s$^{-1}$** | **47 min - 20 h** | **Wave packets and solitary waves (inc fronts, outflow boundaries, wake low); New York and Toronto Metro areas** |
| Grivet-Talocia et al. (1999) | Pressure sensors | 626 | 0.2-10 hPa | 40-600 km (96% of cases) | 0.5-6 h | 5-65 m s$^{-1}$ | *hours to days | Wave packets and solitary waves (inc surface boundaries); Central Illinois |
| Uccellini and Koch (1987) | Pressure sensors | 13 | 0.2-7.0 hPa | 50-500 km | 1-4 h | 13-50 m s$^{-1}$ | 9-33 h | Wave packets and solitary waves; Central and Eastern United States |
| Koch and Siedlarz (1999) | Pressure sensors | 13 | 0.2-0.7 hPa | 200-260 km (3 strongest cases) | 1-6 h | 19-9-27.9 m s$^{-1}$ (3 strongest cases) | 15-38 h | Wave packets and solitary waves; Central United States |
| Bosart et al. (1998) | Pressure sensors | 1 | < 1 hPa strengthening to > 10 hPa | 200-300 km | **1-3 h | 30-40 m s$^{-1}$ | > 18 h | Wave packet with a strong solitary wave; Northeast United States |

**Table 2.** Properties of pressure waves presented in this study (indicated in **bold**) and in some prior studies and meta-analyses in the literature. Rows are in ascending order by minimum wavelength. Information in cells indicated with * were inferred from figures in the associated paper (not directly stated), and the cell indicated with ** was inferred from the wavelength and phase speed range for that paper.



| | Event Start UTC | Event End UTC | Location | Event Type | $N_{sensors}$ | Wave Period mm:ss | Mean Amp. hPa | Phase Speed $\mathrm{m\,s^{-1}}$ | Phase Dir. Degrees | Wavelength km |
|---|---|---|---|---|---|---|---|---|---|---|
| 1 | 2020-11-15 1859 | 2020-11-15 2216 | TOR | Front | 5 | 02:08 | 1.83 | 27.5 | 65.2 | 3.5 |
| **2** | **2020-12-25 0709** | **2020-12-25 1120** | **TOR** | **Gravity wave** | **4** | **26:33** | **1.69** | **48.7** | **29.6** | **77.6** |
| **3** | **2020-12-28 0437** | **2020-12-28 0525** | **TOR** | **Gravity wave** | **4** | **03:21** | **0.45** | **21.6** | **84.7** | **4.3** |
| **4** | **2021-02-18 1418** | **2021-02-18 1642** | **TOR** | **Gravity wave** | **4** | **04:53** | **0.89** | **62.0** | **64.6** | **18.2** |
| **5** | **2021-03-31 0543** | **2021-03-31 0642** | **TOR** | **Gravity wave** | **4** | **03:36** | **0.71** | **25.6** | **14.6** | **5.5** |
| **6** | **2021-04-28 1213** | **2021-04-28 1656** | **TOR** | **Gravity wave** | **4** | **04:34** | **2.73** | **29.6** | **94.8** | **8.1** |
| **7** | **2021-05-01 2140** | **2021-05-02 0051** | **TOR** | **Gravity wave** | **4** | **13:04** | **2.36** | **28.4** | **138.7** | **22.2** |
| 8 | 2021-09-07 1931 | 2021-09-08 0528 | TOR | Outflow | 6 | 08:59 | 3.26 | 21.1 | 120.3 | 11.4 |
| **9** | **2022-01-27 1415** | **2022-01-27 1533** | **TOR** | **Gravity wave** | **4** | **05:05** | **0.78** | **23.5** | **113.3** | **7.2** |
| **10** | **2022-02-25 0816** | **2022-02-25 1359** | **TOR** | **Gravity wave** | **4** | **20:22** | **2.14** | **45.4** | **72.8** | **55.5** |
| **11** | **2022-03-07 0352** | **2022-03-07 0630** | **TOR** | **Gravity wave** | **4** | **32:39** | **1.13** | **52.0** | **33.5** | **101.8** |
| **12** | **2022-03-07 1221** | **2022-03-07 2237** | **TOR** | **Gravity wave** | **4** | **55:42** | **3.09** | **50.7** | **90.3** | **169.4** |
| **13** | **2022-03-30 2118** | **2022-03-31 0133** | **TOR** | **Gravity wave** | **5** | **16:39** | **2.41** | **28.2** | **81.2** | **28.1** |
| **14** | **2022-05-21 1110** | **2022-05-21 1231** | **TOR** | **Gravity wave** | **4** | **05:51** | **0.81** | **30.6** | **104.3** | **10.7** |
| 15 | 2022-05-21 1316 | 2022-05-21 22:47 | TOR | Outflow | 5 | 02:27 | 5.51 | 33.2 | 70.0 | 4.9 |
| **16** | **2023-02-15 0550** | **2023-02-15 0749** | **TOR** | **Gravity wave** | **4** | **08:38** | **1.10** | **24.9** | **53.3** | **12.9** |
| **17** | **2023-02-19 1106** | **2023-02-19 1316** | **TOR** | **Gravity wave** | **5** | **03:23** | **1.20** | **29.0** | **68.8** | **5.9** |
| **18** | **2023-02-23 0139** | **2023-02-23 0323** | **TOR** | **Gravity wave** | **5** | **12:53** | **0.74** | **42.2** | **74.2** | **32.6** |
| **19** | **2023-04-01 0357** | **2023-04-01 0740** | **TOR** | **Gravity wave** | **6** | **21:21** | **1.33** | **37.9** | **48.9** | **48.5** |
| 20 | 2023-04-01 0736 | 2023-04-01 1107 | TOR | Front | 6 | 07:35 | 1.63 | 36.8 | 56.1 | 16.8 |
| 21 | 2023-04-05 0745 | 2023-04-05 1401 | TOR | Outflow | 4 | 05:35 | 3.78 | 27.9 | 90.9 | 9.4 |
| **22** | **2023-04-05 0809** | **2023-04-06 0400** | **TOR** | **Gravity wave** | **6** | **05:29** | **4.24** | **22.1** | **111.8** | **7.3** |
| 23 | 2023-04-16 2250 | 2023-04-17 0206 | TOR | Front | 5 | 04:44 | 1.92 | 19.9 | 12.6 | 5.7 |
| **24** | **2020-01-25 1544** | **2020-01-25 2254** | **NY** | **Gravity wave** | **4** | **09:49** | **2.87** | **18.7** | **52.9** | **11.0** |
| **25** | **2020-02-04 0824** | **2020-02-04 1200** | **NY** | **Gravity wave** | **4** | **16:31** | **1.48** | **17.2** | **179.4** | **17.0** |
| **26** | **2020-05-01 0435** | **2020-05-01 0939** | **NY** | **Gravity wave** | **5** | **30:02** | **2.43** | **19.4** | **64.3** | **34.9** |
| **27** | **2020-12-25 1629** | **2020-12-25 1952** | **NY** | **Gravity wave** | **4** | **04:32** | **1.18** | **47.3** | **39.7** | **12.9** |
| **28** | **2021-02-18 1556** | **2021-02-18 2247** | **NY** | **Gravity wave** | **6** | **60:28** | **2.25** | **32.7** | **114.6** | **118.7** |
| 29 | 2021-09-14 0004 | 2021-09-14 0629 | NY | Wake low | 4 | 66:37 | 3.25 | 20.8 | 68.2 | 83.2 |
| **30** | **2021-12-29 0755** | **2021-12-29 1258** | **NY** | **Gravity wave** | **6** | **11:55** | **1.92** | **33.5** | **130.2** | **24.0** |
| 31 | 2022-02-04 1642 | 2022-02-04 2033 | NY | Outflow | 5 | 14:49 | 1.84 | 21.1 | 117.7 | 18.7 |
| 32 | 2022-02-18 1056 | 2022-02-18 1350 | NY | Front | 4 | 03:00 | 1.66 | 20.5 | 122.5 | 3.7 |
| 33 | 2022-03-08 0119 | 2022-03-08 0428 | NY | Front | 4 | 04:15 | 1.72 | 23.1 | 123.5 | 5.9 |

**Table 3.** Properties of the 33 pressure wave events detected over the 40 month analysis period. The leftmost column is an index column. In the location column, TOR indicates the Toronto sensor network, and NY indicates the New York and Long Island sensor network. The wave period and amplitude are averaged among sensors which detected a given event. Phase direction is shown in degrees clockwise from northward (e.g., 90 degrees indicates a wave propagating west to east). Rows in **bold** indicate gravity wave events.



| | Event start (UTC) | Warm/cold sector | Low-relative position | 300 hPa context |
|---|---|---|---|---|
| 2 | 2020-12-25 0709 | Cold sector | West of low | Downstream of trough |
| 3 | 2020-12-28 0437 | Unclear | East of low | Downstream of trough |
| 4 | 2021-02-18 1418 | Cold sector | North of low | Downstream of trough |
| 5 | 2021-03-31 0543 | Warm sector | South of low | Downstream of trough |
| 6 | 2021-04-28 1213 | Cold sector | Low-adjacent | Near ridge axis |
| 7 | 2021-05-01 2140 | Unclear | No closed low | Upstream of trough |
| 9 | 2022-01-27 1415 | Unclear | South of low | Zonal flow |
| 10 | 2022-02-25 0816 | Cold sector | North of low | Downstream of trough |
| 11 | 2022-03-07 0352 | Cold sector | Northeast of low | Zonal flow |
| 12 | 2022-03-07 1221 | Cold sector | North of low | Downstream of trough |
| 13 | 2022-03-30 2118 | Cold sector | East of low | Near ridge axis |
| 14 | 2022-05-21 1110 | Warm sector | No closed low | Downstream of trough |
| 16 | 2023-02-15 0550 | Cold sector | East of low | Downstream of trough |
| 17 | 2023-02-19 1106 | Unclear | Southeast of low | Zonal flow |
| 18 | 2023-02-23 0139 | Cold sector | Northeast of low | Zonal flow |
| 19 | 2023-04-01 0357 | Cold sector | East of low | Near ridge axis |
| 22 | 2023-04-05 0809 | Warm sector | Southeast of low | Downstream of trough |
| 24 | 2020-01-25 1544 | Cold sector | Northeast of low | Downstream of trough |
| 25 | 2020-02-04 0824 | Cold sector | East of low (far) | Zonal flow |
| 26 | 2020-05-01 0435 | Cold sector | Between lows | Downstream of trough |
| 27 | 2020-12-25 1629 | Cold sector | South of low | Downstream of trough |
| 28 | 2021-02-18 1556 | Cold sector | North of low | Downstream of trough |
| 30 | 2021-12-29 0755 | Cold sector | East of low (far) | Zonal flow |

**Table 4.** Environmental context for the 23 gravity wave events detected over the 40 month analysis period. The leftmost column is an index column, aligned with the index column in Table 3. Here, the position of wave events relative to the low and to air mass boundaries was determined based on manual analysis of $\theta_e$ and MSLP maps derived from ERA5 data at the center time of the event (Fig. 8). If no air mass boundary could be discerned near Toronto or New York for an event, then we consider it "Unclear" whether that event occurred in the cold sector or warm sector. Events 25 and 30 occurred roughly 2000 km or more to the east of the nearest cyclone, as indicated in the table and the text.





| | Event Start | Echo Present | Surface Snow | Surface Rain | Reflectivity Band(s) Present | Reflectivity Band(s) Collocated | Doppler Velocity Wave(s) Present | Doppler Velocity Wave(s) Collocated |
|---|---|---|---|---|---|---|---|---|
| 2 | 2020-12-25 0709 | Yes | Yes | No | No | No | Yes | Yes |
| 3 | 2020-12-28 0437 | Yes | No | Yes | No | No | Yes | No |
| 4 | 2021-02-18 1418 | Yes | Yes | No | No | No | Yes | No |
| 5 | 2021-03-31 0543 | Yes | No | Yes | No | No | No | No |
| 6 | 2021-04-28 1213 | No | No | No | No | No | No | No |
| 7 | 2021-05-01 2140 | Yes | No | Yes | No | No | Yes | No |
| 9 | 2022-01-27 1415 | No | No | No | No | No | No | No |
| 10 | 2022-02-25 0816 | Yes | Yes | No | No | No | Yes | No |
| 11 | 2022-03-07 0352 | No | No | No | No | No | No | No |
| 12 | 2022-03-07 1221 | Yes | Yes | No | Yes | No | Yes | No |
| 13 | 2022-03-30 2118 | Yes | No | Yes | No | No | No | No |
| 14 | 2022-05-21 1110 | No | No | No | No | No | No | No |
| 16 | 2023-02-15 0550 | Yes | No | Yes | No | No | No | No |
| 17 | 2023-02-19 1106 | No | No | No | No | No | No | No |
| 18 | 2023-02-23 0139 | Yes | Yes | No | No | No | Yes | No |
| 19 | 2023-04-01 0357 | Yes | No | Yes | Yes | No | Yes | Yes |
| 22 | 2023-04-05 0809 | Yes | No | Yes | Yes | Yes | Yes | No |
| 24 | 2020-01-25 1544 | Yes | No | Yes | Yes | Yes | Yes | Yes |
| 25 | 2020-02-04 0824 | Yes | No | Yes | No | No | No | No |
| 26 | 2020-05-01 0435 | Yes | No | Yes | Yes | Yes | Yes | No |
| 27 | 2020-12-25 1629 | Yes | No | Yes | No | No | No | No |
| 28 | 2021-02-18 1556 | Yes | Yes | No | Yes | No | Yes | Yes |
| 30 | 2021-12-29 0755 | Yes | No | Yes | No | No | Yes | Yes |
| Total Yes | | 18 | 6 | 12 | 6 | 3 | 13 | 5 |

**Table 5.** Radar and precipitation context for the 23 gravity wave events detected during our 40 month analysis period. The leftmost column is an index column, aligned with the index column in Table 3. The presence of surface snow was determined using the nearest available ASOS data (CYYZ or KJFK). Echo, reflectivity bands, and Doppler velocity waves are considered "present" when they exist anywhere within the range of the $0.5°$ scan for the nearest NEXRAD radar (KBUF or KOKX). Reflectivity bands and Doppler velocity waves are considered "collocated" when they are located directly above pressure sensors and their movement is consistent with the gravity wave phase velocity..



| Toronto | Hours with snow | Hours with other precip | Hours with no precip | Total |
| --- | --- | --- | --- | --- |
| Hours with gravity wave events | 15 | 23 | 48 | 86 |
| Hours without gravity wave events | 445 | 1172 | 16214 | 17831 |
| Total | 460 | 1195 | 16262 | 17917 |
| | | | | |
| New York | Hours with snow | Hours with other precip | Hours with no precip | Total |
| Hours with gravity wave events | 4 | 20 | 13 | 37 |
| Hours without gravity wave events | 130 | 1308 | 16538 | 17976 |
| Total | 134 | 1328 | 16551 | 18013 |

**Table 6.** Hours with and without gravity wave events subdivided by ASOS precipitation data during the November to May months between January 2020 and April 2023. Precipitation is determined to be present when there was $\geq 0.1 \; \mathrm{mm\,hr^{-1}}$ liquid equivalent precipitation was recorded at Toronto (CYTZ precipitation type and CXTO precipitation amount) and at New York (KJFK precipitation type and amount). Hours with either only snow or a mixture of precipitation types containing snow are included under hours with snow.