# Peer review of "Objectively identified mesoscale surface air pressure waves in the context of winter storm environments and radar reflectivity features: a 3+ year analysis"

_EGUsphere, 2024_

## Referee Comment (RC2)

**Paper summary and overview**

This paper explores the role of atmospheric gravity waves during north-east USA winter storms in the absence of orographic influences. Networks of high precision pressure sensors have been positioned in Toronto, ON, Canada and New York, NY, USA, and pressure wave events are identified from the 3+ year timeseries of data that has been collected. Using the meteorological context provided by ERA5, Doppler radar, surface stations and radiosondes, these events are characterised and studied. Consistency with previous literature is found regarding strong upper-level flow imbalance to the south or west of the gravity wave event locations, however not many events are detected with both gravity waves and enhanced snowfall radar bands, and no conclusive relationship between the two can be established from this study.

Overall, this is a really good paper which provides a thorough exploration of non-orographic gravity wave-driven enhancements to north-east USA winter storms and their associated snowfall. The method and theory sections demonstrate that a very detailed experimental process has been followed, and the figures and text have been produced with a lot of care and attention to detail. The scientific findings are a little inconclusive in places, however, agreement with existing literature is found and the results appear to be relatively robust. I can therefore recommend publication in ACP once the following minor comments have been addressed.

**General Comments**

1. I feel as though the abstract doesn't tell the full story about the final conclusions of this paper. There is not much about the conclusion of a lack of a common association between reflectivity bands and gravity waves, and between Doppler velocity waves and gravity waves – which to me forms a large part of the purpose of the study. It also finishes quite abruptly, can this be rounded off in a better way?

2. Given that many of the cases are quite different in both their pressure trace morphology and synoptic context, and with the relatively small sample sizes of similar events when the radar data is taken into account, I am unsure about how concrete any conclusions about the **general** role of gravity waves in winter storms **globally** are from this study. Although much of the method and theoretical work is very good, section 3 seems to finish with a bit of a dead-end and no demonstrable link between gravity waves and snowfall bands, despite the build-up in the introduction section?

3. Some of the background on primary and multi-bands is barely mentioned later on in the results, is this because this aspect was not explored further or because the sample size of relevant events was too small to study this?

4. I wonder whether a slightly looser constraint on defining what is and isn't a gravity wave event could increase the sample size without affecting the results too much, and enable a broader study to be done, even if the uncertainty associated with each individual event is higher?

5. In spite of the above, the science that has been conducted seems to be of a high quality, and even a result which is in agreement with existing literature, or one of limited conclusiveness on the role of gravity waves in winter storm snowfall bands, is still a good scientific outcome for this paper.

**Specific Comments**

1. P01L00: Can the title be shortened and/or simplified to make it more attractive to read? Also, gravity waves (GWs) are not in the title and yet they are front and centre of the start of the abstract. This is intriguing!

2. P03L56: Would it be possible for excessive riming to occur such that ice particles become heavy enough to fall below this region into a warmer/drier layer, such as occurs with some cirrus streaks? I am not sure myself, but wanted to raise the question, as this would be another possible mechanism that rimed ice masses could be removed from the upward branch of a gravity wave. Perhaps the wording "will not be removed" is a little bit strong? Reading on, I can see that the final sentence in the paragraph validates the possibility for removal mechanisms other than the two given in this sentence.

3. P04L95: The range of 5 min to 2 h is quite a large one, I wonder whether there is much difference between primary bands and multibands here that would point towards a greater importance of predicting one type over the other for severe snowfall impacts? I'm assuming that primary bands typically stay over one location for a longer period of time? Is there a difference in the typical intensity of each type, or are they about the same?

4. P05L129: This statement is not true, in observations there are many other ways to definitively confirm the presence of gravity waves, from satellite data to radiosonde profiles, ground-based radar and lidar and aircraft measurements etc. Furthermore, the presence of a perturbation in pressure sensor data is not enough by itself to signify the presence of a gravity wave, as the authors correctly point out elsewhere.

5. P06L153: I wonder if there is a slightly clearer way of writing these three sentences (from "A scale-dependent threshold function..." to "...Mean wavelet power increases with wave period.")? In Allen et al. (2024d, AMT), the necessity for the scale-dependent threshold K is fully explained - with lower values leading to more waves being detected but with the possibility of artefacts, and a K value of 10 leading to only the strongest wave signals being identified. Whilst it's of course important to only summarise the details from that paper here, could this part be rephrased so that the reader has a little more context about K?

6. P10L279: This Data and Methods section is very good, and there are no additional changes that I can identify as being required. The key data sets are introduced in an appropriate order, and a good level of detail is provided about the methods used to analyse these in turn. Great!

7. P10L289: These two sentences follow on from each other a little strangely, could this be very slightly rephrased to improve the flow?

8. P11L309: How might the event extraction method lead to the correlation seen in figure 7? I'm intrigued to know why there was more residual wavelet signal extending beyond the given event duration for the synthetic events that you tested against.

9. P12L355: I wonder if it is a bit unclear to say that 13 (57%) occurred north or east of a surface low, when events 25 and 30 were also to the east? Do you mean that there are 13 that are *close-by* to the north or east (i.e. not 25 and 30 which are further away), and in only *simple* cases (i.e. not 6 or 26 which are more complex)?

10. P13L387: I did find myself getting a little bit lost in section 3.1.2 (from L354 to L372) trying to follow certain events through the text, although I can see the challenges in presenting the information clearly when so many of the cases are unique in various ways. L373 to L387 are much clearer, and the questions raised regarding the surface lows and upper-level troughs earlier on in the subsection are well answered from L380 to L387.

**Figure and Table Comments**

1. General: Please could you stick to one format for labelling colorbars and figures? Sometimes you state the variable with units alongside in brackets (e.g. figure 7), sometimes it is just the units (e.g. figure 8), sometimes the units are in square brackets (e.g. figure 4). The fonts and font sizes are sometimes different as well, can these be unified where this is possible? Please also label subplot panels with letters where possible (a), (b), (c) etc. See the ACP style guide for guidance with this.
2. Figure 2: Could you please add lat/lon labels to these figures, even if it is just to 2(a)?
3. Figure 4: Is it possible to remove the blank tick from figure 4d which shows where there is no wave? Either this or explicitly write out "No Wave" or "Not a wave" perhaps.
4. Figure 5: It is currently a little tricky to distinguish between some of the points on figures 5b and 5c, as well as to match up points and error bars. Would it be possible to make these two plots slightly clearer?
5. Figure 6: Some of the text is a little small on this figure, can this be made a bit larger somehow? I'm conscious that there is limited space to do this though.
6. Figure 8: Not a necessity by any means, but if it is possible to remove the outer rectangular border for each subplot, I feel like it would improve this figure (and others). Some indication of lat/lon bounds, even if just on one subplot, or in a new one in the lower-right corner, would be helpful.
7. Figure 11: Please add lat/lon labels.
8. Figure 13: Please label panels with letters and use same headings as in figure 4.
9. Table 1: Is it possible to sort Table 1 alphabetically, chronologically (first use) or otherwise to make it easier to search for the term required? The idea of having this table is a good one though.
10. Table 2: Can the format of Table 2 be reverted to be identical to that of Table 1 for consistency? I acknowledge that this may be changed anyway during publishing, in which case there is no problem.

**Technical Corrections**

1. P01L26: Please either remove brackets from "(up to 67 min)", or give an indication of whether these are short, moderate or large gravity wave periods.
2. P01L28: Replace "those" with "these".
3. P02L37: Please either insert a comma after "2 to 67 min" or place the numerical values in brackets after "spatial" and "time".
4. P02L37: Replace "larger" with "upper" and "smaller" with "lower".
5. P02L43: Insert two commas to separate out the clause ", at least in part,".
6. P02L47: Please insert "also" after "There has", or make a similar change so that the paragraph flows a little better.
7. P04L109: Remove the 'a' before 'more likely'.
8. P04L117: "Section 3.1.3 puts the pressure waves into context of radar-detected features" reads a little strangely, is there a clearer way to phrase this?
9. P04L118: Please write out "Sect. 4" as "Section 4" for consistency.
10. P14L411: Add "to" between "chose" and "categorize".
11. P15L447: Remove the "a" after "suggests", and "associations" should not be plural.

---

## Author Comment (AC1)

**Reviewer comment**

Author response

*Updated text*
* * *
**Reviewer 1:**

**This work builds on previous analysis and techniques to investigate gravity waves using surface observations, radar, and reanalysis. In particular, it goes beyond the previous case study approach to a broader analysis. The manuscript reflects a thorough examination of each event and the relevant processes related to each one. The discussion and interpretation are sound, and any caveats (in the physical interpretation or from the methodology) are clearly identified. The authors point out that a relatively few number of sufficient (with respect to snow rates) amplitude cases occur during this time period and for these locations.**

**The manuscript has clear motivations and implications, is logically organized, is well written, and includes appropriate figures with excellent supplementary information that helps visualization. I do not have any specific comments that should be addressed.**

**Reviewer 2:**

**Paper summary and overview**

**This paper explores the role of atmospheric gravity waves during north-east USA winter storms in the absence of orographic influences. Networks of high precision pressure sensors have been positioned in Toronto, ON, Canada and New York, NY, USA, and pressure wave events are identified from the 3+ year timeseries of data that has been collected. Using the meteorological context provided by ERA5, Doppler radar, surface stations and radiosondes, these events are characterised and studied. Consistency with previous literature is found regarding strong upper-level flow imbalance to the south or west of the gravity wave event locations, however not many events are detected with both gravity waves and enhanced snowfall radar bands, and no conclusive relationship between the two can be established from this study.**

**Overall, this is a really good paper which provides a thorough exploration of non-orographic gravity wave-driven enhancements to north-east USA winter storms and their associated snowfall. The method and theory sections demonstrate that a very detailed experimental process has been followed, and the figures and text have been produced with a lot of care and attention to detail. The scientific findings are a little inconclusive in places, however, agreement with existing literature is found and the results appear to be relatively robust. I can therefore recommend publication in ACP once the following minor comments have been addressed.**

**General Comments**

**1. I feel as though the abstract doesn't tell the full story about the final conclusions of this paper. There is not much about the conclusion of a lack of a common association between reflectivity bands and gravity waves, and between Doppler velocity waves and gravity waves – which to me forms a large part of the purpose of the study. It also finishes quite abruptly, can this be rounded o in a better way?**

The abstract has been rewritten to address this comment.

**2. Given that many of the cases are quite different in both their pressure trace morphology and synoptic context, and with the relatively small sample sizes of similar events when the radar data is taken into account, I am unsure about how concrete any conclusions about the general role of gravity waves in winter storms globally are from this study. Although much of the method and theoretical work is very good, section 3 seems to finish with a bit of a dead-end and no demonstrable link between gravity waves and snowfall bands, despite the build-up in the introduction section?**

We have tried to clarify in the introduction that, while past studies have demonstrated a connection between gravity waves and heavy snow events, the question we aim to address is *how commonly that association exists* for winter storms in the northeast US and southern Canada. The lack of a demonstrable link in our analysis and the small sample of gravity wave events we were able to find suggest that it is uncommon for gravity waves to be associated with snow bands. We have added this sentence to the first paragraph in Section 1.2:

*Given that past studies such as Bosart et al. (1998) have focused on individual cases where gravity waves were associated with heavy snowfall, there are remaining questions regarding how common that association is for typical winter storms in the northeast United States and southern Canada.*

We also added the words "and how often" to the first sentence in Section 1.4:

*In this study, we used high-precision surface pressure sensors to objectively identify pressure wave events over a 3+ year period, characterize the wave properties and their synoptic environments, and examine whether and how often the pressure waves are related to enhancements in radar reflectivity and coherent sets of Doppler velocity waves.*

Also we have reworded the paragraph in the Conclusions to clarify.

*A few of our detected pressure wave events were associated with frontal passages (5), outflows (4), and a wake low (1), and the remaining 23 were gravity waves, 20 of which occurred in the cool season between November and April. For context, there were 20 snow storms in New York City and 59 in the Toronto metropolitan areas over our 40 month observation period. While limited, the observational evidence we have suggests a lack of a common association between reflectivity bands and gravity waves. Just 6 of the gravity wave events co-occurred with any surface snowfall (including trace amounts). Only 2 of those 6 events had any enhanced reflectivity bands in the vicinity. The spatial wavelengths of the gravity waves and enhanced reflectivity bands were similar, but in all the cases with snow, the reflectivity bands were either not directly over the pressure sensors or not moving at a velocity consistent with the pressure waves (Table 5). Including events with both rain and snow over the pressure sensor networks, only 5 of 18 gravity wave events also had Doppler velocity waves moving at a velocity consistent with the gravity wave phase velocity. This evidence suggests that most low-level velocity waves are not gravity waves.*

**3. Some of the background on primary and multi-bands is barely mentioned later on in the results, is this because this aspect was not explored further or because the sample size of relevant events was too small to study this?**

The sample size of gravity wave events in snow was indeed too small to explore any association between gravity waves and multibands. The lack of detectable gravity wave events on similar spatiotemporal scales to multibands does likely suggest that the two are

not *commonly* associated. We agree that the background material (particularly Hoban 2016) which motivated this work in the first place should be considered in the results, so we added the following sentence to the end of the 3rd paragraph of Section 3.1.3:

*It had been surmised that gravity waves may often be associated with groups of enhanced reflectivity bands in snow (multibands; Hoban, 2016), but we did not find enough gravity wave events on the typical spatiotemporal scales of multibands during snowfall events over our analysis period to support that notion.*

**4. I wonder whether a slightly looser constraint on defining what is and isn't a gravity wave event could increase the sample size without affecting the results too much, and enable a broader study to be done, even if the uncertainty associated with each individual event is higher?**

Looser constraints would yield more events.We chose our constraints in order to identify robust wave events, which have large enough amplitudes to potentially influence cloud and precipitation processes, and for which we have high confidence in our calculation of the wave phase velocity. In other words, we sought to avoid "chasing noise." Section 3.1 of Allen et al. (2024d) discusses the choice of $K$ = 10 (Eq. 1 in the current paper). Essentially, a lower constraint here would likely lead to the erroneous pairing of separate weak events across pressure sensors which may have ended up in the final analysis.

For high confidence in the calculation of the wave phase velocity, we require RMSE < 90 s and NRMSE < 1 (Eqs. 5-6). These threshold choices came up during the review process for Allen et al. (2024d) as well. We show the RMSE and NRMSE for all wave events detected by at least 4 sensors in Fig. R1 along with the RMSE and NRMSE limits. A small error here means that the lag times between sensors (found using the maximum cross-correlation between the extracted event trace in each sensor) are consistent with each other. We consider errors on the order of hundreds of seconds to be unacceptable, but the exact thresholds we chose are subjective. The 90 s RMSE and 0.1 NRMSE values appear to be a reasonable break in the data, where one or both would have to be increased substantially in order to incorporate many more events in the final analysis.

*Figure R1. NRMSE against RMSE for wave events detected by at least 4 sensors in the Toronto and New York pressure sensor networks. In (a), all wave events are shown. In (b), only wave events with RMSE below 500 s are shown. The RMSE and NRMSE thresholds of 90 s and 0.1, respectively, to consider a wave event trackable are indicated by the purple outline.*

[Figure]

We have changed and added to the beginning of the 3rd paragraph in Section 2.1.1 (line 169), which now reads:

*To identify robust wave signals with large enough amplitudes to potentially modify cloud and precipitation processes, we used K = 10. A lower threshold could lead to the detection of many weak wave events, which may then be erroneously paired across multiple sensors when they were separate wave events in reality.*

We also changed the beginning of the paragraph beginning on line 200 to now read:

*To have reasonable confidence in the wave phase velocity estimate for a given event, we require the event to be captured by at least 4 sensors with RMSE below 90 s and NRMSE below 0.1. These thresholds are based on the analysis in Allen et al. (2024d). We consider RMSE on the order of hundreds of seconds to be unacceptably large, and after analyzing all wave events captured by ≥ 4 sensors, we found RMSE = 90 s and NRMSE = 0.1 to be a break in the data.*

**5. In spite of the above, the science that has been conducted seems to be of a high quality, and even a result which is in agreement with existing literature, or one of limited conclusiveness on the role of gravity waves in winter storm snowfall bands, is still a good scientific outcome for this paper.**

**Specific Comments**

**1. P01L00: Can the title be shortened and/or simplified to make it more attractive to read? Also, gravity waves (GWs) are not in the title and yet they are front and centre of the start of the abstract. This is intriguing!**

A very helpful suggestion, we have revised the title to:

*Hunting for gravity waves in non-orographic winter storms using 3+ years of regional surface air pressure networks and radar observations*

**2. P03L56: Would it be possible for excessive riming to occur such that ice particles become heavy enough to fall below this region into a warmer/drier layer, such as occurs with some cirrus streaks? I am not sure myself, but wanted to raise the question, as this would be another possible mechanism that rimed ice masses could be removed from the upward branch of a gravity wave. Perhaps the wording "will not be removed" is a little bit strong? Reading on, I can see that the final sentence in the paragraph validates the possibility for removal mechanisms other than the two given in this sentence.**

Yes, the process you described would remove ice mass (and overall water mass) from the air parcel following a riming event. This would fall under the process of precipitation fallout, which we do acknowledge as a potential irreversible change to the air parcel resulting from gravity wave uplift. In the sentence referenced here, we are referring to the increased *particle* ice mass by riming only being reversed if $RH_{ice}$ falls below 100%. We changed that sentence to read:

*If lifting associated with a gravity wave brings an ice or mixed phase cloud parcel to liquid water saturation, and if riming then occurs, that rimed ice mass will not be removed from particles unless $RH_{ice}$ falls below 100% in the downward branch of the gravity wave and sublimation occurs.*

**3. P04L95: The range of 5 min to 2 h is quite a large one, I wonder whether there is much difference between primary bands and multibands here that would point towards a greater importance of predicting one type over the other for severe snowfall impacts? I'm assuming that primary bands typically stay over one location for a longer period of time? Is there a difference in the typical intensity of each type, or are they about the same?**

Intensity (i.e., snowfall rate) does not monotonically increase with reflectivity for snow, so gauging the intensity of a band from reflectivity alone is not possible (Tomkins 2024, Ph.D. dissertation). To our knowledge, nobody has compared snowfall rates associated with primary bands to those associated with multibands. The duration which an enhanced reflectivity band stays over a fixed location is important to the severity of snowfall impacts. The range of durations given here is based on an assumption of 10-30 m s$^{-1}$ band propagation speeds in the band-perpendicular direction. If the band motion has some band-parallel component, the duration which the band stays over a fixed location could be even longer, especially for primary bands. We have changed this paragraph (now 4th paragraph in Section 1.3, Line 105) to read:

*The snowfall accumulation associated with a reflectivity band is a function of the intensity (i.e., snowfall rate) and duration that the band is over a fixed location. Snowfall rate does not monotonically increase with reflectivity (Fujiyoshi et al., 1990; Rasmussen et al., 2003), so measuring band intensity from radar data alone has large uncertainties. Reflectivity in snow can be increased by aggregation or partial melting of ice particles, which would not increase the associated snow mass. Additionally, localized reflectivity enhancements observed by radar a few km above the surface may not reach the surface (Tomkins, 2024). Given ground-relative propagation speeds on the order of 10-30 m $s^{-1}$ in the band-perpendicular direction, enhanced reflectivity bands usually pass over a given location in 5 to 120 min. Longer durations over a location are possible when there is a band-parallel component to the motion.*

**4. P05L129: This statement is not true, in observations there are many other ways to definitively confirm the presence of gravity waves, from satellite data to radiosonde profiles, ground-based radar and lidar and aircraft measurements etc. Furthermore, the presence of a perturbation in pressure sensor data is not enough by itself to signify the presence of a gravity wave, as the authors correctly point out elsewhere.**

We respectfully disagree. To clarify we have added the following material to the first paragraph of Section 2.1.

*In observations, gravity waves can be implied from the presence of ripples in satellite-observed cloud tops, and in Doppler velocity data observed by radar and lidar (e.g., Miller et al., 2022). But similar ripple-like structures can occur with Kelvin-Helmholtz waves (Houser and Bluestein, 2011). To definitively distinguish between gravity waves and Kelvin-Helmholtz waves, pressure sensor data are needed (e.g., Christie, 1992). Gravity waves will have a pressure wave signature, while Kelvin-Helmholtz waves will not. Not all pressure waves are gravity waves (Allen et al., 2024d).*

**5. P06L153: I wonder if there is a slightly clearer way of writing these three sentences (from "A scale-dependent threshold function…" to "…Mean wavelet power increases with wave period.")? In Allen et al. (2024d, AMT), the necessity for the scale-dependent threshold K is fully explained - with lower values leading to more waves being detected but with the possibility of artefacts, and a K value of 10 leading to only the strongest wave signals being identified. Whilst it's of course important to only summarise the details from that paper here, could this part be rephrased so that the reader has a little more context about K?**

In addition to the changes described under General Comment 4, we have rearranged and added to the material here so that the beginning of the 2nd paragraph in Section 2.1.1 (line 165) explains that the general increase in wavelet power with scale is the reason our threshold function also depends on scale:

*Mean wavelet power increases with scale (i.e., wave period; Allen et al., 2024d, their Fig. 5). Therefore, the threshold defining wave events should also vary with scale.*

**6. P10L279: This Data and Methods section is very good, and there are no additional changes that I can identify as being required. The key data sets are introduced in an appropriate order, and a good level of detail is provided about the methods used to analyse these in turn. Great!**

**7. P10L289: These two sentences follow on from each other a little strangely, could this be very slightly rephrased to improve the flow?**

From previous version

285     There did not appear to be a strong relationship between wave period and wave amplitude for pressure wave events (Fig. 5b), which is somewhat surprising, given that the mean wavelet power generally increases with wave period for pressure (Canavero and Einaudi, 1987; Grivet-Talocia and Einaudi, 1998; Allen et al., 2024d). Individual pressure wave events (Fig. 5) may not follow the same pattern of increasing amplitude with increasing wave period as seen in longer-term mean values of wavelet power (Allen et al., 2024d, their Fig. 5). The pressure wave events are caused by atypical short-term pressure perturbations

290     whereas the long-term mean wavelet power mainly consists of quiescent conditions, usually without sharp pressure changes. Figure 5b includes the range of wave periods where the wavelet power exceeded $A(a)$ as error bars. From these error bars, it is apparent that nearly every pressure wave event had a strong wave signal at shorter wave periods ($< 30$ min), while very few had a strong wave signal at longer wave periods ($> 90$ min).

We trimmed out what was likely the confusing portion so it now reads as (now on Line 304):

*There did not appear to be a strong relationship between wave period and wave amplitude for pressure wave events (Fig. 5b), which is somewhat surprising, given that the mean wavelet power generally increases with wave period for pressure (Canavero and Einaudi, 1987; Grivet-Talocia and Einaudi, 1998; Allen et al., 2024d). Figure 5b includes the range of wave periods where the wavelet power exceeded A(a) as error bars. From these error bars, it is apparent that nearly every pressure wave event had a strong wave signal at shorter wave periods (< 30 min), while very few had a strong wave signal at longer wave periods (> 90 min).*

**8. P11L309: How might the event extraction method lead to the correlation seen in figure 7? I'm intrigued to know why there was more residual wavelet signal extending beyond the given event duration for the synthetic events that you tested against.**

One of the synthetic wave events (Allen et al. 2024d, Section 3.1) is shown in Figure R2. The synthetic wave event had a prescribed duration of 120 minutes. From the middle panel (wavelet power), it is apparent that times beyond the prescribed duration of the wave event still had some wavelet signal. At, e.g., 7:20 UTC (10 minutes before the prescribed wave event started) and a wave period of 15 minutes, the wavelet power still exceeded 5 times the mean wavelet power for that wave period, so that time was included in the *detected*

wave event. Specifically, the detected wave event had a duration of 162 minutes, 42 minutes longer than the prescribed duration. For synthetic wave events of a higher amplitude, the "extra" duration in the detected wave event is longer. For synthetic wave events with a constant 2 hour prescribed duration, amplitudes ranging from 0.01 hPa to 1 hPa, and wave periods ranging from 2 min to 120 min, we found that the detected event duration increases by roughly 88 minutes per 1 hPa increase in detected wave amplitude (Fig. R3), which is substantially less than the 180 minute increase per 1 hPa increase in amplitude for the wave events detected from real pressure data (Fig. 7). So, the extraction method might be a part of the reason we found a correlation between gravity wave event duration and amplitude, but it is likely only a small part if it contributes at all.

*Figure R2. (Top) Synthetic pressure (hPa) time series created by summing a constant pressure (randomly chosen from normal distribution with mean 1000 hPa and standard deviation 2 hPa) with normally distributed random noise (standard deviation 0.008 hPa, equal to the pressure sensor noise floor) and a synthetic sine wave event. The synthetic wave event starts at 7:30 UTC, ends at 9:30 UTC, has a peak amplitude of ±1 hPa, and has a wave period of ~25 minutes. The amplitude ramps up and ramps down linearly during the first and last 12 minutes of the wave event. (Middle) Wavelet power (shaded, hPa2 s⁻¹) calculated from the synthetic pressure time series, with contours of normalized wavelet power at 5 (dotted) and 10 (solid). (Bottom) Extracted wave event (hPa) from the synthetic pressure time series.*

[Figure]

*Figure R3. Detected wave event amplitude (hPa) against detected wave event duration (min) for synthetic wave events similar to the one shown in Fig. R2 and as discussed by Allen et al. (2024d). Each wave event had a prescribed duration of 120 min, with amplitudes ranging from ±0.01 to ±1 hPa and wave periods ranging from 2 to 120 min. Detected wave event duration and detected wave event amplitude for these synthetic events have a linear correlation coefficient R = 0.73.*

[Figure]

**9. P12L355: I wonder if it is a bit unclear to say that 13 (57%) occurred north or east of a surface low, when events 25 and 30 were also to the east? Do you mean that there are 13 that are close-by to the north or east (i.e. not 25 and 30 which are further away), and in only simple cases (i.e. not 6 or 26 which are more complex)?**

We have reworded this sentence and the following sentence to try to clarify:

*Of the 23 gravity wave events in Toronto and New York during our 40-month analysis period, 15 (65%) occurred north or east of a surface low (events 3, 4, 10, 11, 12, 13, 16, 17, 18, 19, 22, 24, 25, 28 and 30), often on the cool side of warm or stationary fronts. This includes events 25 and 30, which were more than 2000 km away from the low.*

We also have added to the end of this paragraph so that it now reads:

*The remaining cases are more complex. Event 6 occurred near a weak surface low and just on the cool side of an air mass boundary. Event 26 occurred near a weak air mass boundary with lows both to the north and the south. Events 7 and 14 occurred with no closed low in the region (Fig. 8).*

**10. P13L387: I did find myself getting a little bit lost in section 3.1.2 (from L354 to L372) trying to follow certain events through the text, although I can see the challenges in presenting the information clearly when so many of the cases are unique in various ways. L373 to L387 are much clearer, and the questions raised**

**regarding the surface lows and upper-level troughs earlier on in the subsection are well answered from L380 to L387.**

We have reordered the material in Section 3.1.2 and reworded to improve clarity.

**Figure and Table Comments**

**1. General: Please could you stick to one format for labelling colorbars and figures? Sometimes you state the variable with units alongside in brackets (e.g. figure 7), sometimes it is just the units (e.g. figure 8), sometimes the units are in square brackets (e.g. figure 4). The fonts and font sizes are sometimes different as well, can these be unified where this is possible? Please also label subplot panels with letters where possible (a), (b), (c) etc. See the ACP style guide for guidance with this.**

We have removed the unit labels from the colorbar to the title for Figures 8, 9, 10, and 11, to be consistent with Figures 4, 13, and 14. In Figure 12, we left the unit label in the colorbar as the primary focus of the figure is on the height of inversion layers and the lapse rates are secondary. We have also changed the square brackets around units to parentheses in Figures 4, 13, and 14. We have made the fonts consistent across Figures 8, 9, and 10. Finally, we added letter labels to each panel in Figures 6, 8, 9, and 10.

**2. Figure 2: Could you please add lat/lon labels to these figures, even if it is just to 2(a)?**

We have now added lat/lon gridlines and labels to Figure 2a.

**3. Figure 4: Is it possible to remove the blank tick from figure 4d which shows where there is no wave? Either this or explicitly write out "No Wave" or "Not a wave" perhaps.**

We have added a "No wave" label to Figures 4d, 13d, and 14d.

**4. Figure 5: It is currently a little tricky to distinguish between some of the points on figures 5b and 5c, as well as to match up points and error bars. Would it be possible to make these two plots slightly clearer?**

We have adjusted the size of the symbols used in Figures 5b and 5c, and we have increased the thickness of the error bars in Figure 5c. This mainly reduces the amount of overlap between points in Figure 5c, and we also moved the radial axis labels so that they do not cover any points.

**5. Figure 6: Some of the text is a little small on this figure, can this be made a bit larger somehow? I'm conscious that there is limited space to do this though.**

We have adjusted the font sizes in Figure 6. In particular this should make the axis tick labels easier to read, but there is especially limited space for the panel titles and variable labels.

**6. Figure 8: Not a necessity by any means, but if it is possible to remove the outer rectangular border for each subplot, I feel like it would improve this figure (and others). Some indication of lat/lon bounds, even if just on one subplot, or in a new one in the lower-right corner, would be helpful.**

We have removed the outer rectangular border from each subplot in Figures 8-10, and added a latitude/longitude reference panel to those figures.

**7. Figure 11: Please add lat/lon labels.**

We have added lat/lon labels to Figure 11.

**8. Figure 13: Please label panels with letters and use same headings as in figure 4.**

The panel labels and headings are now consistent for Figures 4, 13, and 14.

**9. Table 1: Is it possible to sort Table 1 alphabetically, chronologically (first use) or otherwise to make it easier to search for the term required? The idea of having this table is a good one though.**

Table 1 is now sorted alphabetically.

**10. Table 2: Can the format of Table 2 be reverted to be identical to that of Table 1 for consistency? I acknowledge that this may be changed anyway during publishing, in which case there is no problem.**

Tables 1 and 2 are now consistent (both have outer border lines).

**Technical Corrections**

Each of these technical corrections has been made.

**1. P01L26: Please either remove brackets from "(up to 67 min)", or give an indication of whether these are short, moderate or large gravity wave periods.**

**2. P01L28: Replace "those" with "these".**

**3. P02L37: Please either insert a comma after "2 to 67 min" or place the numerical values in brackets after "spatial" and "time".**

**4. P02L37: Replace "larger" with "upper" and "smaller" with "lower".**

5. P02L43: Insert two commas to separate out the clause ", at least in part,".

6. P02L47: Please insert "also" after "There has", or make a similar change so that the paragraph flows a little better.

7. P04L109: Remove the 'a' before 'more likely'.

8. P04L117: "Section 3.1.3 puts the pressure waves into context of radar-detected features" reads a little strangely, is there a clearer way to phrase this?

9. P04L118: Please write out "Sect. 4" as "Section 4" for consistency.

10. P14L411: Add "to" between "chose" and "categorize".

11. P15L447: Remove the "a" after "suggests", and "associations" should not be plural.